# The domesticated transposase ALP2 mediates formation of a novel Polycomb protein complex by direct interaction with MSI1, a core subunit of Polycomb Repressive Complex 2 (PRC2)

Christos N. Velanis[1☉], Pumi Perera[1☉], Bennett Thomson[2], Erica de Leau[1], Shih Chieh Liang[1], Ben Hartwig[3], Alexander Förderer[3], Harry Thornton[1], Pedro Arede[1], Jiawen Chen[1], Kimberly M. Webb[4], Serin Gümüs[5], Geert De Jaeger[6,7], Clinton A. Page[8], C. Nathan Hancock[8], Christos Spanos[4], Juri Rappsilber[4,9], Philipp Voigt[4], Franziska Turck[3], Frank Wellmer[2], Justin Goodrich[1]*

**1** Institute of Molecular Plant Science, School of Biological Sciences, University of Edinburgh, Daniel Rutherford Building, Max Born Crescent, Edinburgh, United Kingdom, **2** Smurfit Institute of Genetics, Trinity College Dublin, Ireland, **3** Department of Plant Developmental Biology, Max Planck Institute for Plant Breeding Research, Köln, Germany, **4** Wellcome Centre for Cell Biology, School of Biological Sciences, University of Edinburgh, Max Born Crescent, Edinburgh, United Kingdom, **5** Department of Biotechnology, Mannheim University of Applied Science, Mannheim, Germany, **6** Department of Plant Biotechnology and Bioinformatics, Ghent University, Gent, Belgium, **7** VIB Center for Plant Systems Biology, Gent, Belgium, **8** Department of Biology & Geology, University of South Carolina Aiken, Aiken, South Carolina, United States of America, **9** Bioanalytics Unit, Institute of Biotechnology, Technische Universität Berlin, Berlin, Germany

☉ These authors contributed equally to this work.

* Justin.Goodrich@ed.ac.uk

**Data Availability Statement:** The mass spectrometry proteomics data have been deposited

## Abstract

A large fraction of plant genomes is composed of transposable elements (TE), which provide a potential source of novel genes through "domestication"–the process whereby the proteins encoded by TE diverge in sequence, lose their ability to catalyse transposition and instead acquire novel functions for their hosts. In Arabidopsis, ANTAGONIST OF LIKE HETEROCHROMATIN PROTEIN 1 (ALP1) arose by domestication of the nuclease component of *Harbinger* class TE and acquired a new function as a component of POLYCOMB REPRESSIVE COMPLEX 2 (PRC2), a histone H3K27me3 methyltransferase involved in regulation of host genes and in some cases TE. It was not clear how ALP1 associated with PRC2, nor what the functional consequence was. Here, we identify ALP2 genetically as a suppressor of Polycomb-group (PcG) mutant phenotypes and show that it arose from the second, DNA binding component of *Harbinger* transposases. Molecular analysis of PcG compromised backgrounds reveals that *ALP* genes oppose silencing and H3K27me3 deposition at key PcG target genes. Proteomic analysis reveals that ALP1 and ALP2 are components of a variant PRC2 complex that contains the four core components but lacks plant-specific accessory components such as the H3K27me3 reader LIKE HETEROCHROMATION PROTEIN 1 (LHP1). We show that the N-terminus of ALP2 interacts directly with ALP1, whereas the C-terminus of ALP2 interacts with MULTICOPY SUPPRESSOR OF

to the ProteomeXchange Consortium via the
PRIDE partner repository (http://proteomecentral.
proteomexchange.org/cgi/GetDataset) Dataset
Identifier: PXD018911.

**Funding:** This publication has emanated from
research supported in part by a research grant
from Science Foundation Ireland (SFI) under the
Grant Number SFI/16/BBSRC/3392 and the BBSRC
under the Grant Reference BB/P008569/1 and
funded JG, FW, CV, EdL, PA and BT. Preliminary
work for this project was supported by the Gatsby
Charitable Foundation (PP) and by a Darwin Trust
studentship (SCL). PP was supported by a UK
Gatsby Trust studentship, SCL by a Darwin Trust of
Edinburgh studentship. PV was supported by the
Wellcome Trust [104175/Z/14/Z, Sir Henry Dale
Fellowship]. The Wellcome Centre for Cell Biology
was supported by core funding from the Wellcome
Trust [203149]. The funders had no role in study
design, data collection and analysis, decision to
publish, or preparation of the manuscript.

**Competing interests:** The authors have declared
that no competing interests exist.

IRA1 (MSI1), a core component of PRC2. Proteomic analysis reveals that in *alp2* mutant
backgrounds ALP1 protein no longer associates with PRC2, consistent with a role for ALP2
in recruitment of ALP1. We suggest that the propensity of *Harbinger* TE to insert in gene-
rich regions of the genome, together with the modular two component nature of their trans-
posases, has predisposed them for domestication and incorporation into chromatin modify-
ing complexes.

## Author summary

A large part of the genomes of plants and animals consists of transposable elements (TE),
which are usually considered as selfish or parasitic as they encode proteins (transposases)
which promote TE proliferation but not functions useful for their hosts. As a result, hosts
have evolved ways of reducing TE proliferation, usually by modifying the DNA or chro-
matin of TE so that their transposases are no longer produced. Once the TE are inacti-
vated they can no longer proliferate and over time they accumulate mutations and can
evolve new functions, often beneficial for their hosts. This process is known as domestica-
tion and is increasingly recognised as a potent source of evolutionary novelty. For exam-
ple, the CRISPR/Cas system that has provided the basis for a revolution in genetic
engineering ("genome editing") has evolved via domestication of transposons in bacteria.
We have identified the ALP proteins, two domesticated transposases which function as
components of an enzyme complex (PRC2) involved in modifying chromatin and regulat-
ing host gene activity in plants. Here we show how ALPs contact PRC2 and direct forma-
tion of a novel complex that lacks several of the usual components. The ALPs and related
proteins will provide valuable tools for manipulating plant chromatin.

## Introduction

A large fraction of plant and animal genomes is made up of transposable elements (TE), and
even in plant species such as *Arabidopsis thaliana*, which has a relatively small genome, 15% of
the genome is comprised of TE [1]. Unconstrained proliferation of TE is deleterious for their
hosts, as insertions can disrupt genes and also promote genome rearrangements including
translocations, inversions, chromosome breakage and loss. As a result, hosts have consistently
evolved surveillance mechanisms that monitor and inhibit TE activity. Typically, these involve
hosts producing small RNA molecules with homology to TE transcripts, leading to the local
recruitment of repressive epigenetic machinery to TE chromatin and ultimately to the silenc-
ing of the genes encoding the transposase proteins that catalyse transposition and proliferation
[for reviews see 2, 3]. In principle this could lead to an evolutionary arms race, in which TE
evolve proteins that target and inhibit components of the host defence machinery, as has been
well documented for plant viruses and pathogens [for review see 4]. However, an important
distinction is that TE, unlike viruses and pathogens, are propagated vertically by replication
and transmission through the germline of their hosts. Unconstrained TE activity that reduces
host fitness will therefore be equally catastrophic for the propagation and persistence of the
TE. As a result, rather than an arms race scenario, a détente model is more appropriate, in
which TE have co-evolved with their hosts and adopted mutualistic strategies that limit the
cost of their propagation to their hosts and which lead to the domestication of TE to serve host
specific functions [5]. One common way in which TE can benefit hosts is when insertion of a

TE nearby a gene introduces regulatory sequences that alter the expression of the gene in a way that is advantageous. For example, many traits that have been selected during crop domestication involve regulatory changes caused by TE insertions, including anthocyanin production in blood oranges [6], low fruit acidity in citrus [7] and reduced branching in maize [8]. Ultimately this can lead to domestication as the regulatory sequences are maintained but not the ability to transpose. A second, perhaps less well-recognised way in which TE contribute to host genome evolution is as a rich source of novel genes which arise by domestication of TE encoded genes. Domestication here refers to the process in which transposase genes acquire mutations such that their products (transposases, reverse transcriptases, integrases, coat proteins etc.) no longer function in catalysing transposition and instead evolve novel functions. Domesticated transposases have been identified both from forward genetic screens based on phenotypes, but increasingly from bioinformatic analysis of genome sequences; in the latter, identification is based on a suite of common properties including sequence similarity with functional transposases, low copy number, a conserved genomic location in different species (immobilisation), detectable expression, and a lack of TE-associated features such as terminal inverted repeats [for review see 9]. For example, these features allowed identification of 36 novel genes putatively derived from domesticated transposases in Arabidopsis, and there are likely many more that escaped detection because their sequences have diverged sufficiently that similarities to transposases are no longer apparent [10]. Similar approaches have identified domesticated transposases in vertebrates and flies [11, 12] but in most cases the function of these genes is unknown, raising the question as to whether each has acquired a different role or whether there are common functions.

*Harbinger* class TE, also termed *PIF* and *Pong* in plants, encode two separate proteins, one with DNA binding activity and the other a nuclease, a distinctive feature in that most other DNA class TE encode a single transposase with both functions [12, 13]. The two proteins are both needed for transposition and can interact with one another, reconstituting transposase activity [14–16]. A genetic screen for mutations affecting silencing of a transgene in Arabidopsis identified two genes that oppose silencing and evolved by domestication of *Harbinger* genes: one, *HARBINGER DERIVED PROTEIN 1* (*HDP1*) is related to the nuclease component, whilst the other, *HARBINGER DERIVED PROTEIN 2* (*HDP2*) is related to the DNA binding component. Proteomic studies revealed that the HDP1 and HDP2 proteins are associated with the INCREASED DNA METHYLATION (IDM) complex, one member of which (IDM1) is a histone acetyltransferase that potentially catalyses acetylation associated with gene activation [17]. A second genetic screen, aimed at identifying factors that oppose silencing mediated by Polycomb group (PcG) genes, identified *ANTAGONIST OF LIKE HETEROCHROMATIN PROTEIN 1* (*ALP1*) and showed that it encoded another domesticated *Harbinger*-related nuclease [18]. ALP1 was also identified bioinformatically as one of 36 high confidence domesticated TE genes [10]. Proteomic studies showed that, consistent with its genetic interactions with the PcG, ALP1 is associated with the PcG protein complex POLYCOMB REPRESSIVE COMPLEX2 (PRC2). The PRC2 is an ancient H3K27me3 histone methyltransferase that was first identified genetically in Drosophila through its role in repression of developmental target genes, although in some species or cell types it has also been linked to TE regulation ([19–22]. The canonical PRC2 contains four core components which in Arabidopsis are mostly represented by small gene families [for review see 23]–for example, the catalytic subunit, Enhancer of zeste (E[z]) in Drosophila, is represented by three homologues in Arabidopsis, CURLY LEAF (CLF), SWINGER (SWN) and MEDEA (MEA). In many species the canonical PRC2 is associated with accessory components which can modify the activity of the complex or provide additional functions such as DNA binding specificity [24–28]. The accessory components are less conserved and usually specific to particular lineages. For example, in plants

immunoprecipitation experiments indicate that the core PRC2 is associated with several plant-specific PcG proteins that share common mutant phenotypes with the PRC2 members. Thus in Arabidopsis, various proteomic studies show that the chromodomain-containing protein LIKE HETEROCHROMATIN PROTEIN 1 (LHP1)/TERMINAL FLOWER 2 (TFL2), is a PRC2 component [18, 26, 29]. LHP1 is homologous to HETEROCHROMATIN PROTEIN 1 (HP1) of animals, but its function has diverged as it binds H3K27me3 unlike HP1 which binds the heterochromatic mark H3K9me2. EMBRYONIC FLOWER1 (EMF1) is a plant-specific PcG protein whose function is less clear, but biochemical studies suggest it may play a role in chromatin compaction and transcriptional silencing [30]. Genomics studies indicate that EMF1 is bound to chromatin at many PcG target genes and that it is required for normal H3K27me3 levels at many (44%) of these targets, suggesting that it may be linked to PRC2 activity [31]. Consistent with this, several proteomic studies indicate that EMF1 associates with PRC2 [18, 29], although additional studies suggest it may also participate in several other PcG complexes [32, 33]. Lastly, the plant-specific PHD domain protein VERNALIZATION INSENSITIVE 3 (VIN3) and three closely related proteins VIN3-LIKE 1–3 (VIL1-3, also known as VRN5, VEL1 and VEL2) are components of variant PRC2 complexes [18, 34].

A striking feature of the ALP1 containing PRC2 complex is that it lacks accessory components such as LHP1 or EMF1 [18]. However, it is not clear how ALP1 interacts with the PRC2 nor whether there are additional components of the ALP-PRC2 complex. Here, we identify an additional member of this complex, *ANTAGONIST OF LIKE HETEROCHROMATIN PROTEIN1 2* (*ALP2*), and show that–like HDP1 and HDP2 –ALP1 and ALP2 were co-domesticated from the nuclease and DNA binding components of *Harbinger* transposases, respectively. The ALP2 protein interacts directly with both ALP1 and PRC2 and is required to recruit ALP1 to the PRC2. We suggest that ALP2 may compete with accessory factors for access to the PRC2, altering the activity or targeting specificity of the ALP-PRC2 complex.

## Results

### *ALP2* acts in the same genetic pathway as *ALP1*

We previously described a genetic screen for modifiers of PcG phenotypes in which we identified the *ALP1* gene [35]. In the same screen, we identified another mutant which similar to *alp1* gave a partial suppression of the narrow, curled leaf and early flowering phenotype of the *lhp1* mutant (Fig 1A and 1B). Genetic analysis showed that it corresponded to a recessive mutation in a separate gene which we designated *ANTAGONIST OF LIKE HETEROCHRO-MATIN PROTEIN 2* (*ALP2*). Similar to *alp1*, *alp2* mutations also partially suppressed the clf mutant phenotype (Fig 1C). These similarities suggested that *ALP1* and *ALP2* act in a common genetic pathway. Consistent with this, the *alp1-1 alp2-1 lhp1-3* triple mutant did not give a greater suppression of the lhp1 phenotype than did the *alp2-1 lhp-3* double mutant (Fig 1A). For example, *lhp1* mutants are early flowering and this is partially suppressed by *alp* mutations, so that the *lhp1-3 alp2-1* double mutants flower later than *lhp1-3* single mutants but earlier than the wild type (Fig 1B). However, the *lhp1-3 alp1-1 alp2-1* triple mutants did not flower later than either *lhp1-3 alp1-1* or *lhp1-3 alp2-1* double mutants (Fig 1B).

To characterise the *lhp1 alp* phenotypes further, we measured expression of *AGAMOUS* (*AG*), *SEPALLATA3* (*SEP3*) and *FLOWERING LOCUS T* (*FT*), three PRC2 target genes whose mis-expression in PcG mutants causes the early flowering, leaf curling phenotype [36, 37]. All three target genes were strongly up-regulated in *lhp1* mutant seedlings grown in short days (Fig 2A). The expression of *SEP3* and *FT* was much reduced in *lhp1 alp* double mutants, consistent with the suppression of the lhp1 phenotype. In contrast, there was less effect on *AG* expression as was previously found for *clf-50 alp1-4* mutants [18]. In the *lhp1 alp1 alp2* triple

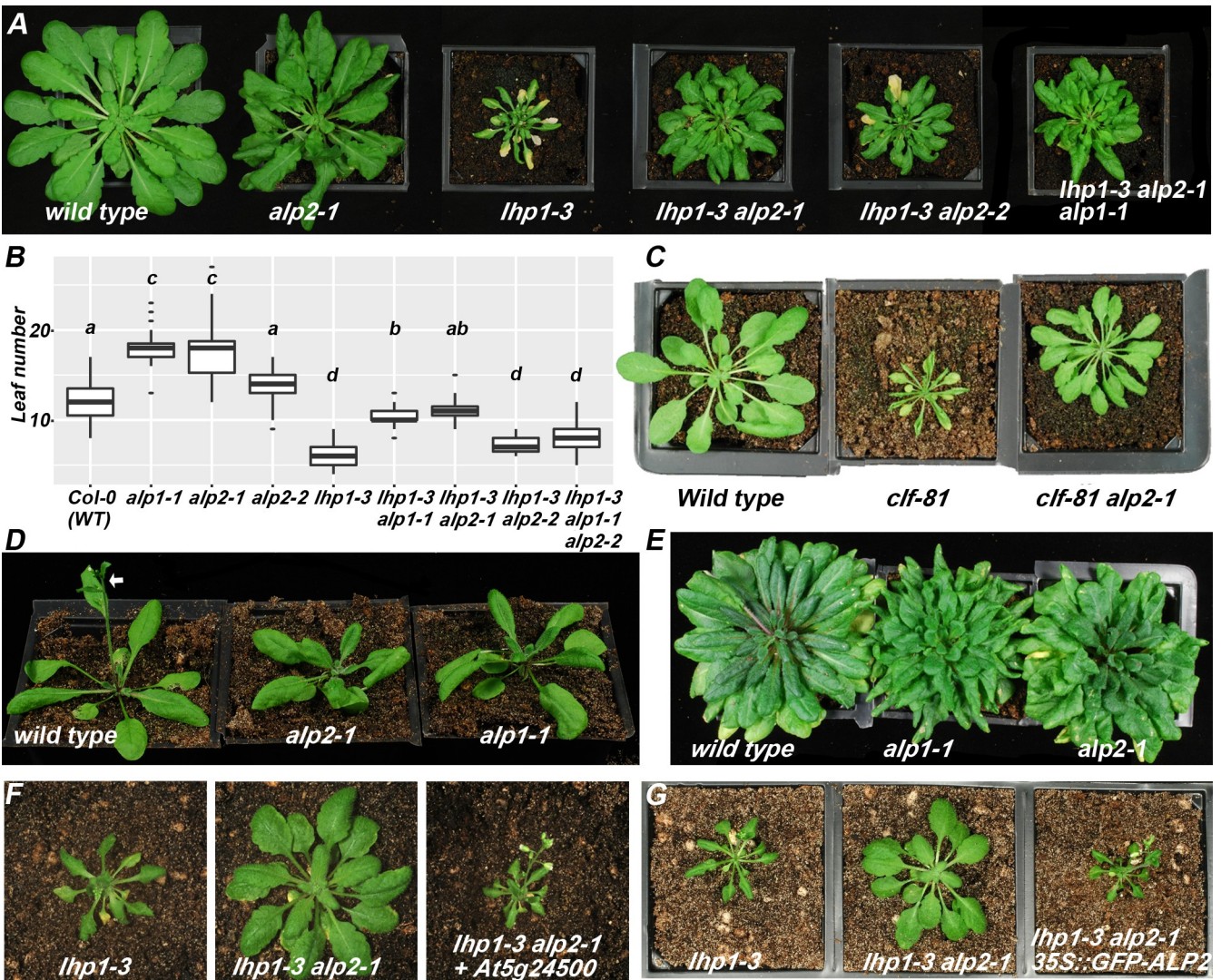

**Fig 1. *ALP2* acts in the same pathway as *ALP1*.** A. Rosette phenotypes of 9 week old plants grown in short days. B. Box plots showing median and inner quartiles of flowering time data (number of rosette leaves) of plants grown in long days. Whiskers delimit largest value within 1.5 times of inner quartiles. Statistical analysis was performed with analysis of variance (ANOVA) and Holm-Sidak multiple-testing correction. Different letters indicate distinct groups (two-tailed P values <0.05) C. Suppression of clf phenotype by *alp2-1*. D. *alp1-1* and *alp2-1* single mutants flower later than wild-type (Col-0) in long days but do not show gross phenotypic abnormalities. White arrow indicates inflorescence in Col-0, plants are four weeks old. E. *alp2-1* and *alp1-1* plant grown in short days show altered rosette morphology. F. Complementation of the alp2 mutant phenotype by an *At5g24500* transgene. G. Complementation of the alp2 mutant phenotype by a *35S::GFP-ALP2* transgene.

mutant plants there was no further reduction in *SEP3* or *FT* expression relative to the *lhp1 alp* double mutants. Together these results showed that *alp1* and *alp2* have similar mutant phenotypes and that in the absence of *ALP1* activity there was no further effect of withdrawing *ALP2* activity, so *ALP1* and *ALP2* act in a common genetic pathway. Consistent with this, *ALP2* was widely expressed in plant tissues similar to *ALP1* and *CLF* (Fig 2B).

Although the *alp* mutations had a strong effect in PcG mutant backgrounds, their phenotypes were weaker in wild-type (*LHP1*[+]) backgrounds. However, both *alp1-1* and *alp2-1* mutations were slightly late flowering relative to their wild-type (Col-0) progenitors (Fig 1B and 1D) and had mild vegetative phenotypes in which older leaves were curled slightly downward

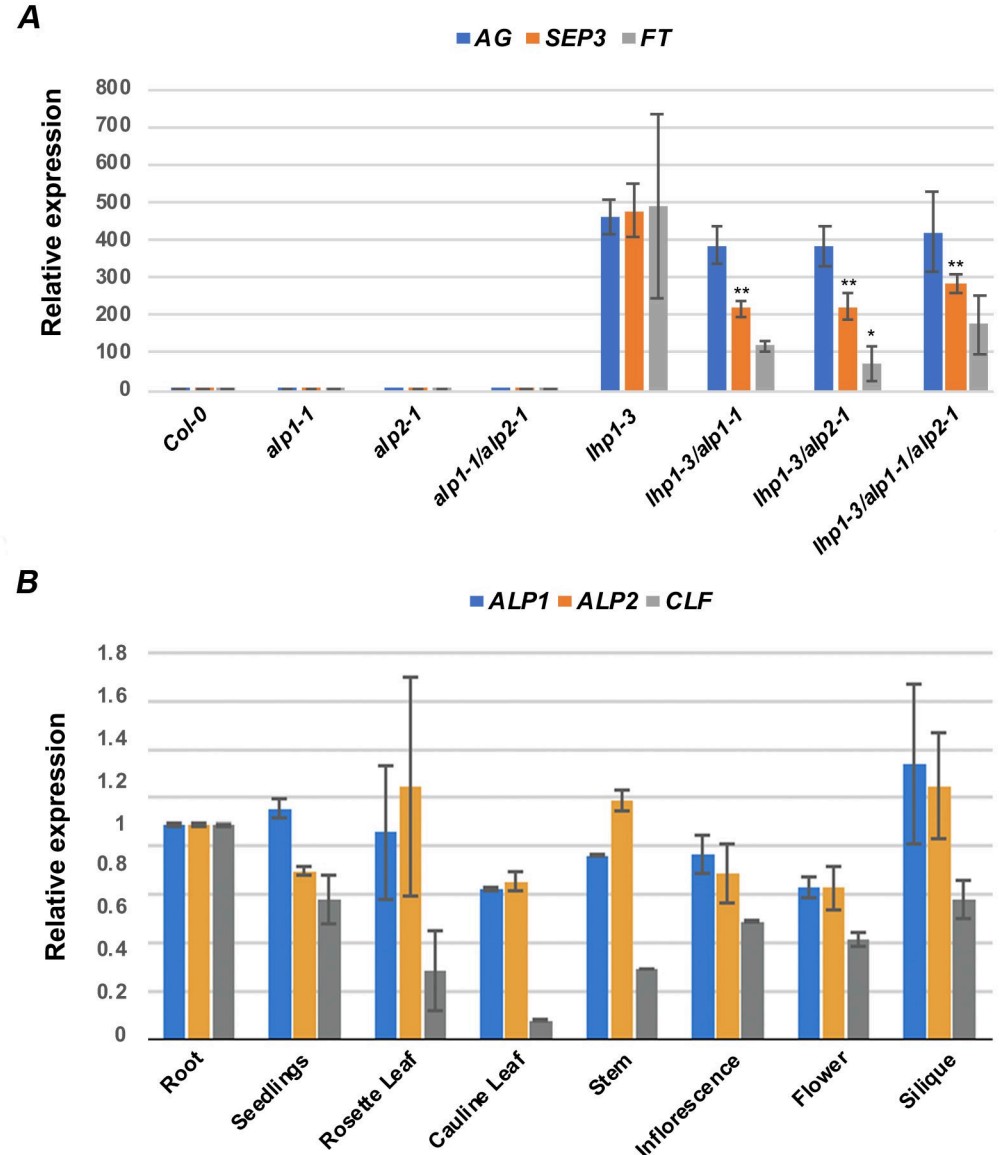

**Fig 2. Expression of PcG targets and *ALP* genes.** A. Real time RT PCR analysis of *AG*, *SEP3*, and *FT* expression in 12 day old seedlings grown in short days. Expression for each gene was normalised to the reference gene *EiF4A1*. Values are shown relative to the level in wild type (Col-0) seedlings (set to 1) and mean and standard deviation of three biological replicates. Asterisks indicate *lhp alp* sample means significantly different from *lhp1-1* (** p < .01, * p < .05, Tukey's pairwise comparisons). B. Expression of *ALP1*, *ALP2* and *CLF* in different plant organs normalised to *EiF4A1*. Expression is shown relative to levels in roots, set to 1, and shows mean and standard deviation of two biological replicates.

(Fig 1E). To further characterize the effects of the *ALP* genes on floral induction, we grew plants in non-inductive short days and then transferred plants to long days and measured the expression of the floral homeotic genes *PISTILLATA* (*PI*), *AG* and *SEP3* in shoot apices at different times after the shift. Both *alp1* and *alp2* showed reduced expression relative to the wild type (S1 Fig) in agreement with a role of the *ALP* genes in the photoperiodic induction of flowering.

## *ALP2* encodes a novel protein with similarity to *Harbinger* transposase DNA binding proteins

To identify the *ALP2* gene we used the mapping by sequencing approach described previously [35] and compared DNA from pools of 270 *alp2-1 lhp1* mutant plants with DNA from the *lhp1* parental line. We identified two candidate mutations (S1 Table) but focussed on one, encoding a nonsense mutation W213* in the *At5g24500* product, as we had independently identified this protein in a yeast two hybrid screen for ALP1 interactors (S1 Dataset). To confirm that *ALP2* corresponded to *At5g24500*, we obtained an independent, T-DNA insertion allele (*alp2-2*) and introduced it into the *lhp1* mutant background. Similar to *alp2-1*, we observed a partial suppression of the lhp1 phenotype (Fig 1A and 1B). In addition, when a transgene containing genomic DNA spanning the *At5g24500* locus was introduced into the *alp2-1 lhp1-3* mutant background, the alp2 mutant phenotype was complemented such that the transgenic plants had the ALP2+ (non-suppressed lhp1) phenotype (Fig 1F).

The *ALP2* gene encodes a predicted protein of 334 amino acids that is rich in charged amino acids (35% of all residues), but lacks similarity with any protein of known function. Structural prediction algorithms suggest that ALP2 is predominantly disordered but that residues in the C-terminus (235–261) form a coiled-coil structure. BLAST searches revealed that ALP2 is well conserved throughout angiosperms, and orthologues were also found in a conifer (*Metasequoia glyptostroboides*) and a fern (*Pilularia globulifera*). *Harbinger* TE encode two proteins, a nuclease and a myb domain DNA binding protein, and these interact with one another effectively reconstituting a single, functional transposase [14]. Since ALP1 is related to *Harbinger* nucleases, and ALP1 and ALP2 proteins interact (see below), this raised the possibility that ALP2 might have evolved from a *Harbinger* transposase DNA binding protein. Consistent with this, alignment of plant ALP2 proteins with selected *Harbinger* DNA binding proteins and the domesticated *Harbinger* related protein HDP2 from *A. thaliana* revealed three regions of sequence similarity (Fig 3). These did not include the predicted coiled-coil region but encompassed three blocks of residues that were previously found to be conserved between plant *Harbinger* (*Pong* class) transposase DNA binding proteins [13]. Phylogenetic analysis confirmed that the land plant ALP2 sequences form a single, well supported group distinct from those of HDP2 proteins or *Harbinger* transposase DNA binding domains, consistent with an ancient origin in land plants similar to that of ALP1 (S2 Fig). Notably, all of the land plant ALP2 proteins lacked the three tryptophan residues that are conserved in *Harbinger* DNA binding transposases and the HDP2 domesticates and are required for their DNA binding activity (Fig 3) [11, 17]. ALP2 is therefore unlikely to bind DNA using the Myb domain. Additionally, unlike active transposases, it is present as a single copy gene in most plant genomes, is not located adjacent to a nuclease protein (e.g. ALP1), and is present in syntenic regions between the related species *A. lyrata* and *A. thaliana*, suggesting that it is immobile (S3 Fig). We therefore concluded that ALP2 is a domesticated, highly diverged *Harbinger* transposase DNA binding protein.

## ALP2 associates with the PRC2 complex *in vivo*

Since ALP1 protein associates with the PRC2 complex, we used immunoprecipitation and mass spectrometry (IP-MS) to determine whether ALP2 was also a part of this complex. We first expressed an N-terminal fusion of GFP to ALP2 under control of the constitutive 35S promoter (*35S::GFP-ALP2*). When introduced into the *lhp1-3 alp2-1* mutant background the construct complemented the alp2 mutant phenotype confirming that the GFP-ALP2 fusion protein was functional (Fig 1G). In addition, Western blot analysis of protein extracts confirmed that *35S::GFP-ALP2* expressed a protein with the predicted size for the intact

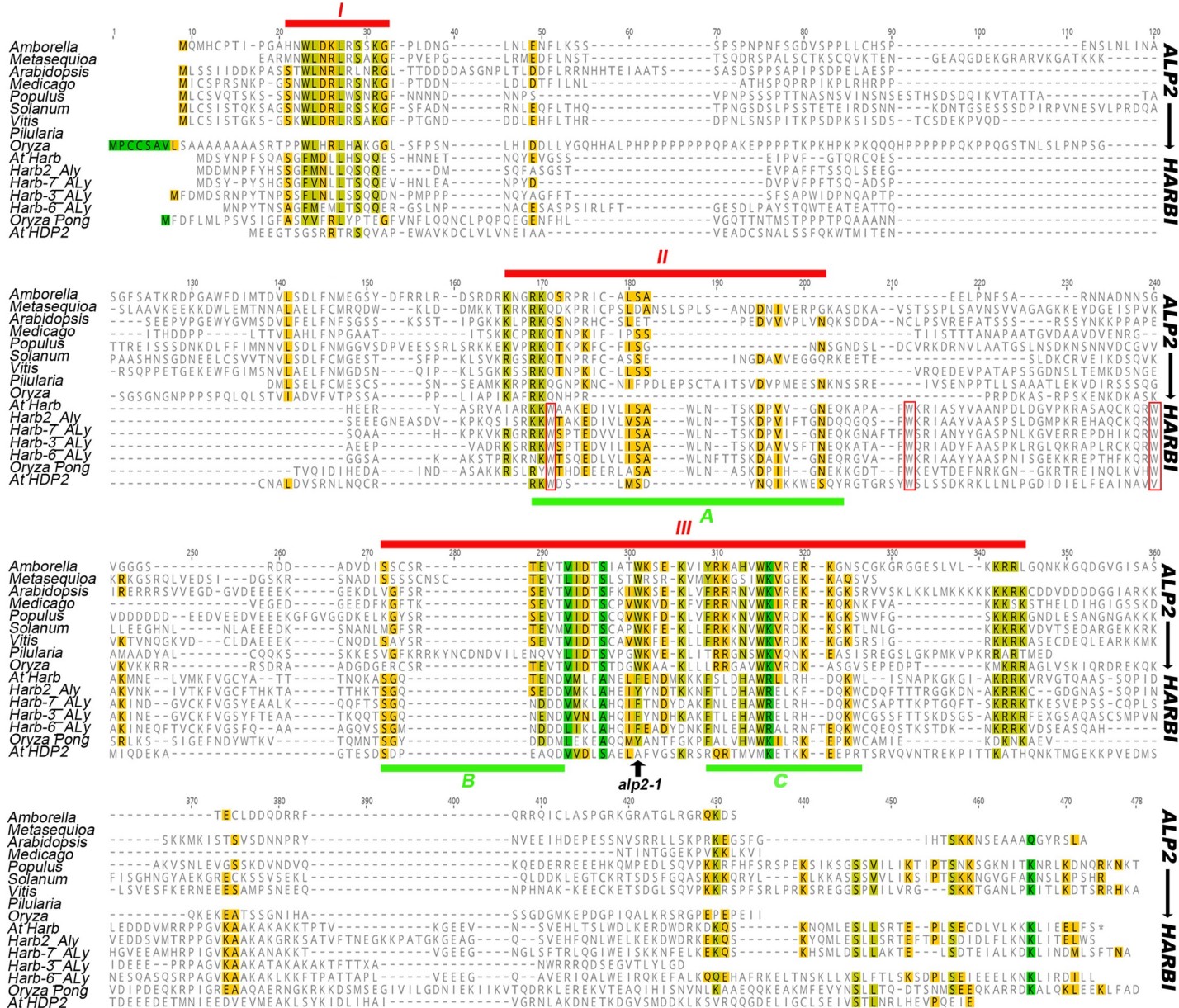

**Fig 3. ALP2 has similarity with *Harbinger* transposase DNA binding proteins.** Alignment using MAFTT of protein sequences of selected land plant ALP2 proteins (upper 8 sequences), *Harbinger* transposase DNA binding proteins and the domesticated transposase HDP2 (lower 9 sequences). The ALP2 sequences include a gymnosperm (*Metasequoia glyptostroboides*) and a leptosporangiate fern (*Pilularia globifera*). Full sequences and accession numbers are provided in S1 Text. Alignment of ALP2 proteins alone identified three regions of sequence conservation (I–III) indicated by the red lines above the alignments, these overlapped with three regions (A, B and C) of conservation between plant *Harbinger* DNA binding proteins (PONG class) previously identified [13] and indicated by green lines below the sequences. The three tryptophan residues that are highly conserved amongst *Harbinger* DNA binding proteins and HDP2 (but not in ALP2 proteins) are outlined with red boxes.

GFP-ALP2 fusion (S4 Fig). We immunoprecipitated protein extracts from inflorescence tissue of *35S::GFP-ALP2*, *pALP1::ALP1-GFP* and *35S::GFP* plants using an antibody to the GFP tag and identified the proteins isolated by mass spectrometry (IP-MS). Consistent with ALP2 interacting with ALP1, we identified numerous ALP1 peptides when ALP2 was immunoprecipitated, although a few (0–3) ALP1 peptides were also found in the *35S::GFP* control; in the ALP1 IP, abundant ALP2 peptides were detected and none were found in the *35S::GFP* control

**Table 1. ALP2 associates with ALP1 and the PRC2 core components.** Results from three independent immunoprecipitation experiments. The numbers indicate the number of uniquely identified peptides from each protein found by mass spectrometry. The total number of peptides identified in each experiment is also indicated. PRC2 core components are highlighted in dark blue, accessory components in light blue. The full list of proteins identified is presented as an excel sheet in S2 Table.

| Protein | 35S::GFP | p(ALP1)::ALP1-GFP | p(ALP1)::ALP1-GFP alp2 | 35S::GFP-ALP2 |
|---------|----------|-------------------|------------------------|---------------|
| ALP1 | 0-3-4 | 27-28-30 | 28-27-20 | 9-12-12 |
| ALP2 | 0-0-0 | 12-12-10 | 1-0-0 | 18-15-23 |
| EMF2 | 0-0-0 | 22-22-22 | 2-0-0 | 26-24-28 |
| MSI1 | 0-0-0 | 22-21-21 | 5-0-1 | 25-21-25 |
| FIE | 0-0-0 | 16-20-20 | 1-0-0 | 18-15-18 |
| SWN | 0-0-0 | 29-38-37 | 0-0-0 | 26-24-38 |
| CLF | 0-0-0 | 16-9-12 | 0-0-0 | 22-17-24 |
| EMF1 | 0-0-0 | 0-0-0 | 0-0-0 | 0-0-0 |
| LHP1 | 0-0-0 | 0-0-0 | 0-0-0 | 0-0-0 |
| VRN5 | 0-0-0 | 0-0-0 | 0-0-0 | 0-0-0 |
| VEL1 | 0-0-0 | 0-0-0 | 0-0-0 | 0-0-0 |
| ALL PEPTIDES | 3052-5266-6486 | 10875-12712-11537 | 11986-7056-5409 | 7399-8495-8872 |

(Table 1 and S2 Dataset). Strikingly, the ALP2 IP clearly contains all four core members of the Arabidopsis PRC2 (i.e., FIE, MSI1, EMF2, SWN/CLF) (Table 1). As with the ALP1 IP, ALP2 did not pull down any of the accessory components (LHP1, EMF1, VRN5, VEL1) that have been found when the PRC2 core components are immunoprecipitated [18, 26, 29]. To identify which proteins were most strongly and reliably enriched in our IP-MS experiments we also performed a "volcano" plot analysis, in which we quantified the relative abundance of each protein identified (relative to control samples) and plotted this against the statistical significance of each enrichment (t test p value from three independent experiments). This confirmed that in the ALP1 and ALP2 IP experiments, the core PRC2 components were amongst the most highly enriched and reproducibly identified proteins (Fig 4A and 4B and S3 Dataset).

Since the above IP-MS was all done using floral tissue and GFP-tagged proteins, as an independent validation we expressed ALP2 in a different cell type (dark grown Arabidopsis suspension culture cells) as a fusion with the GS$^{rhino}$ tandem tag comprising two protein G tags and a Streptavidin binding peptide separated by the rhinovirus 3C protease cleavage sites [38]. Following a two-step affinity purification of extracts from 35S::GS$^{rhino}$-ALP2 transgenic cells and control (non-transgenic) cells in three replicate experiments, we analysed the eluates by MS. Again, numerous peptides from ALP1 and the core PRC2 members specifically co-purified with ALP2 and were amongst the most highly enriched and statistically robust proteins in a volcano plot analysis (S2 Table and S5 Fig). Collectively our proteomic analyses confirm that ALP1 and ALP2 are part of a variant PRC2 complex (ALP-PRC2) that lacks the accessory components LHP1, EMF1, VRN5 and VEL1.

## ALP2 protein interacts directly with ALP1 but not with PING transposase

To test whether the interaction of ALP1 and ALP2 proteins is direct, we cloned the full-length proteins as fusions to the GAL4 DNA binding (bait constructs) or transcriptional activation domains (prey constructs) and performed yeast two hybrid assays. Neither protein had transcriptional activation activity when fused to GAL4 DNA binding domain, but the ALP1 and ALP2 constructs interacted with each other reciprocally as both bait and prey fusions (Fig 5A), consistent with a direct interaction between the two proteins. We next tested the effect of the *alp1-1* missense mutation (G273E) upon the interaction, as this mutation eliminates *ALP1* activity *in vivo* [18]. The *alp1-1* mutation abolished the interaction of ALP1 with ALP2 in yeast

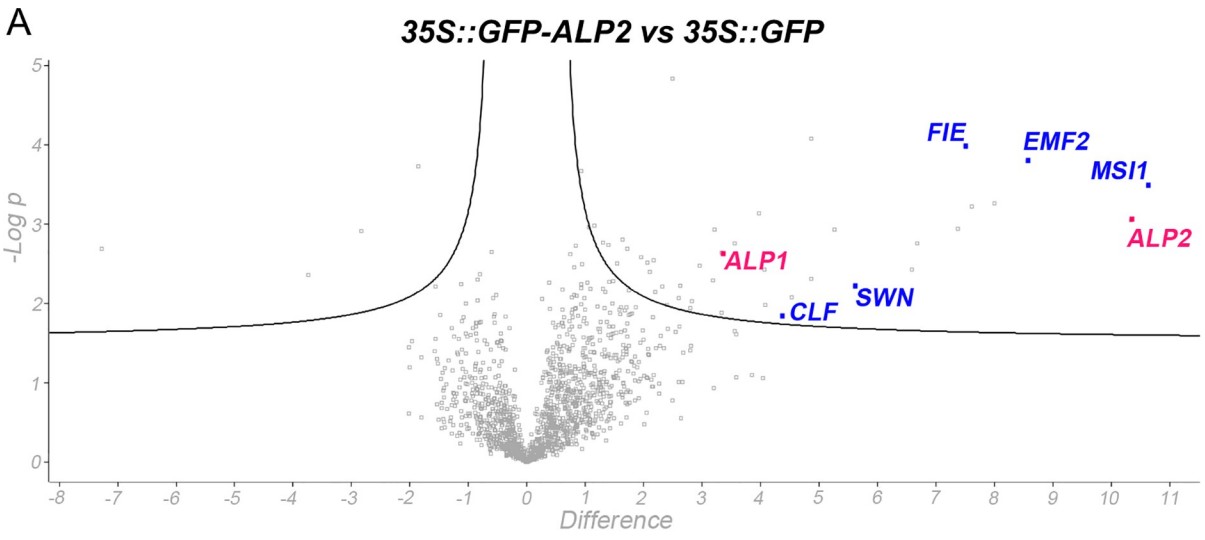

**A** *35S::GFP-ALP2 vs 35S::GFP*

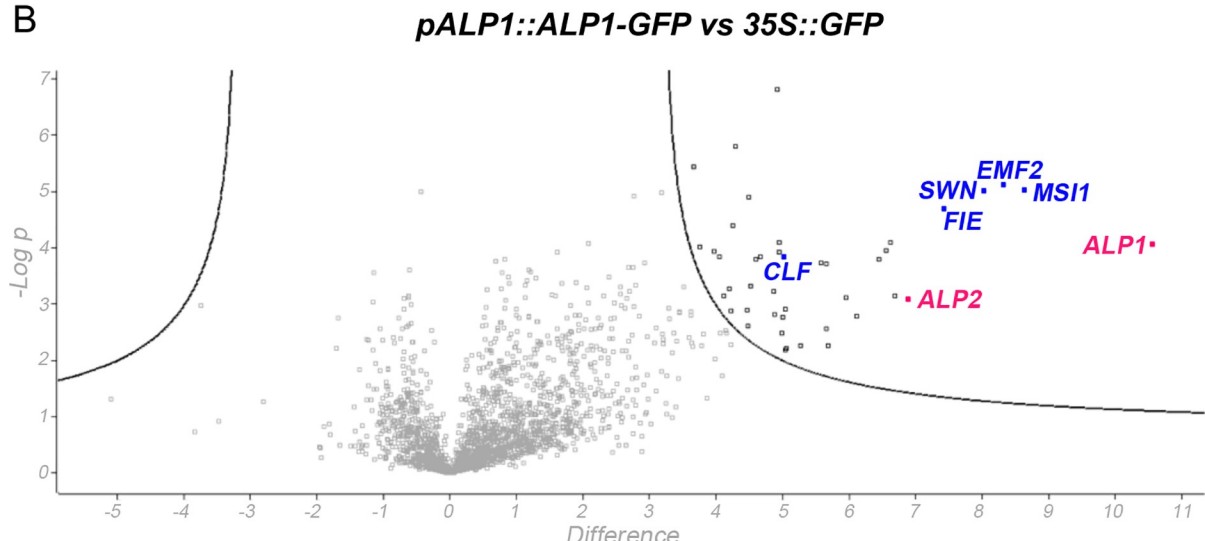

**B** *pALP1::ALP1-GFP vs 35S::GFP*

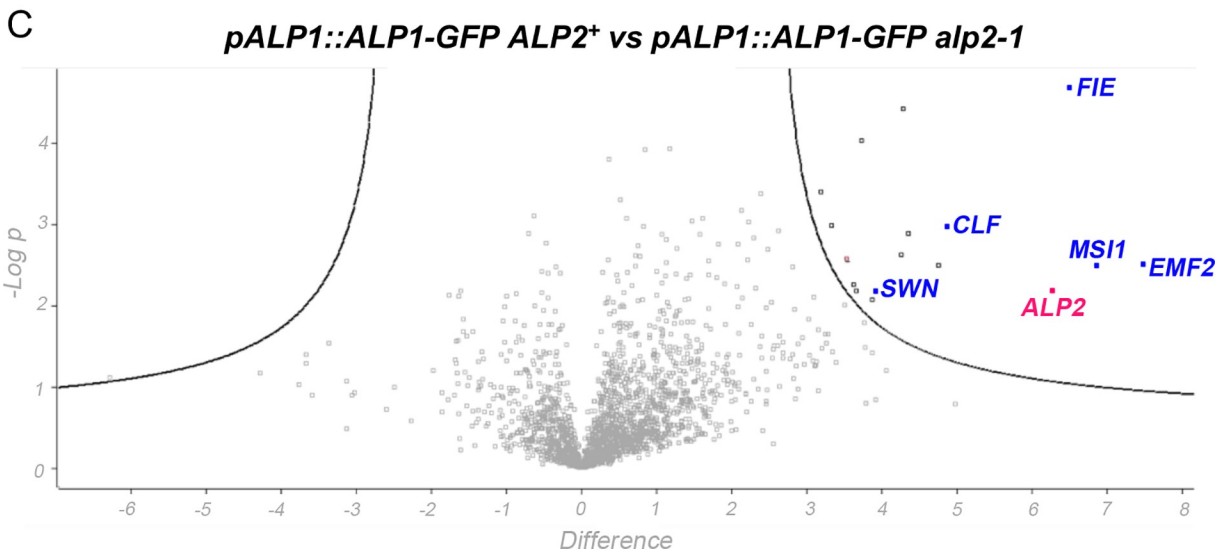

**C** *pALP1::ALP1-GFP ALP2⁺ vs pALP1::ALP1-GFP alp2-1*

**Fig 4. Volcano plot analysis of IP-MS data.** The relative abundance of proteins compared between two samples (log$_2$ of fold difference, x axis) is plotted against statistical significance (-log$_{10}$ of *P*-value, y axis) with each point representing a protein identified by 2 or more unique peptides. The curved lines on right hand side of plots delimit outliers with high relative abundance and significance in bait sample. PRC2 components are highlighted in blue, ALPs in red, for the full list of the most highly enriched proteins see S3 Dataset. A. Comparison of *35S:: GFP-ALP2* and *35S:GFP* IPs shows that ALP1 and PRC2 core components are highly enriched. B. Comparison of *pALP1::ALP1-GFP* with *35S:: GFP* IPs shows that ALP2 and PRC2 core components are highly enriched. C. Comparison of *pALP1::ALP1-GFP* IPs in *ALP2$^+$* and *alp2-1* backgrounds shows that PRC2 core components are highly enriched in *ALP2$^+$* samples and depleted in *alp2-1*.

(Fig 5A). Using protein truncations, we identified a C-terminal portion (amino acids 171–261) of ALP2 that spanned a well conserved region (III in Fig 3) and was sufficient for interaction with ALP1 (Fig 5B). Furthermore, the *alp2-1* mutation (W213*) deletes most of this conserved region (Fig 3), suggesting that it is important for ALP2 activity *in vivo*. Neither N-terminal or C-terminal parts (residues 1–200 and 190–396, respectively) of ALP1 were sufficient for interaction with ALP2 (S6A Fig).

To test whether ALP1 and ALP2 may directly interact in plants, we used the bimolecular fluorescence complementation (BiFC) assays in transformed tobacco (*Nicotinia benthamiana*) leaves. We observed strong fluorescence in the nucleus when full length ALP1 and ALP2 proteins were co-expressed as fusions with N- and C-terminal portions of YFP, consistent with ALP1 and ALP2 interacting directly and reconstituting YFP fluorescence. By contrast, no fluorescence was observed in control experiments using fusions with the ALP1-1 (G273E) variant of ALP1 (Fig 6A and S7 Fig). Together these results suggest that the ALP1 and ALP2 proteins interact directly and that the interaction is relevant for their activity *in vivo*.

The ALP1 and ALP2 proteins are related to transposase proteins encoded by *Ping* and *Pong* class TE in plants, raising the question whether the TE proteins interact with one another via

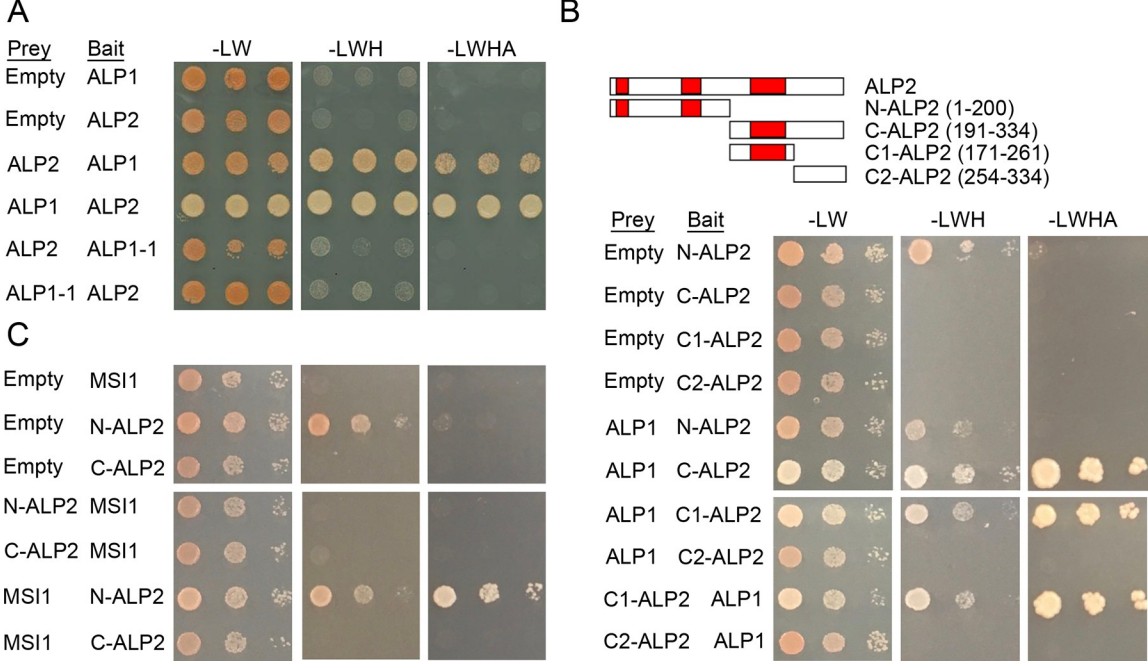

**Fig 5. Interaction of ALP proteins in yeast two hybrid assays.** A. ALP1 and ALP2 protein interaction is abolished by the *alp1-1* (G273E) missense mutation. B. A C-terminal region of ALP2 containing a conserved region is sufficient for the interaction with ALP1. C. The N-terminus of ALP2 interacts with MSI1. Although the N-ALP2 bait alone shows weak autoactivation ability (growth on -LWH), in the presence of MSI prey the more stringent ADE reporter is activated, allowing growth on -LWHA medium. In panel A, three independent transformants are shown, in B and C serial ten-fold dilutions of five pooled transformants.

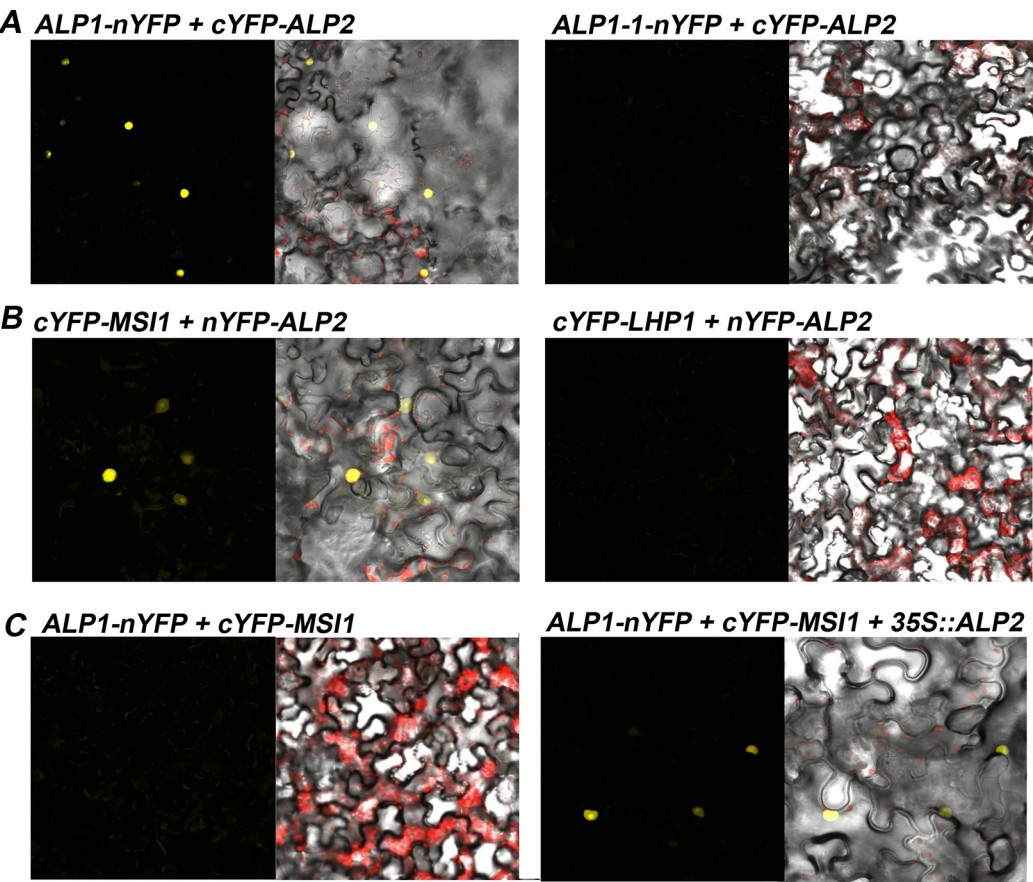

**Fig 6. Interactions of ALP proteins in BiFC assay.** A. ALP2 interacts with ALP1 but not with ALP1-1. B. ALP2 interacts with MSI1 but not with LHP1. C. ALP1 does not interact with MSI1 unless ALP2 is co-expressed. *N. benthamiana* leaves were transformed by infiltration with Agrobacterium and the epidermis viewed using confocal microscope. Images on the left show the YFP channel, and on the right a merged light field (showing cell outlines) and fluorescence channels. Chloroplasts present in stomata or underlying cell layers are autofluorescent and appear red. Low magnification images, showing larger numbers of cells, and further controls are shown in S5 Fig.

similar domains and if so whether they can also interact with ALPs. To address this we performed yeast two hybrid assays using the Myb DNA binding protein and the nuclease protein (also termed ORF1 and TPase, respectively) encoded by the rice *Ping* TE. Consistent with the fact that both Ping proteins are needed to catalyse transposition, we found that their full length proteins interacted in yeast (S8 Fig). To delimit the interacting regions of the Ping proteins further, we made truncations and found that a N-terminal half of the *Ping* nuclease was sufficient to interact with full length Myb DNA binding protein, whereas the C-terminal half of the Myb DNA binding protein interacted with the full length nuclease when expressed as a prey but not a bait fusion (S9 Fig). However, ALP2 did not interact with *Ping* nuclease and ALP1 did not interact with *Ping* Myb, either as full length proteins or in the truncated forms (S8 and S9 Figs).

## ALP2 recruits ALP1 to the PRC2 via MSI1

We next tested for direct interaction of ALP1 and ALP2 with individual PRC2 components using BiFC, yeast two hybrid and pulldown assays. In BiFC assays ALP2 interacted with MSI1 (Fig 6) whereas all other combinations of ALP2 or ALP1 with PRC2 members did not (S7 Fig).

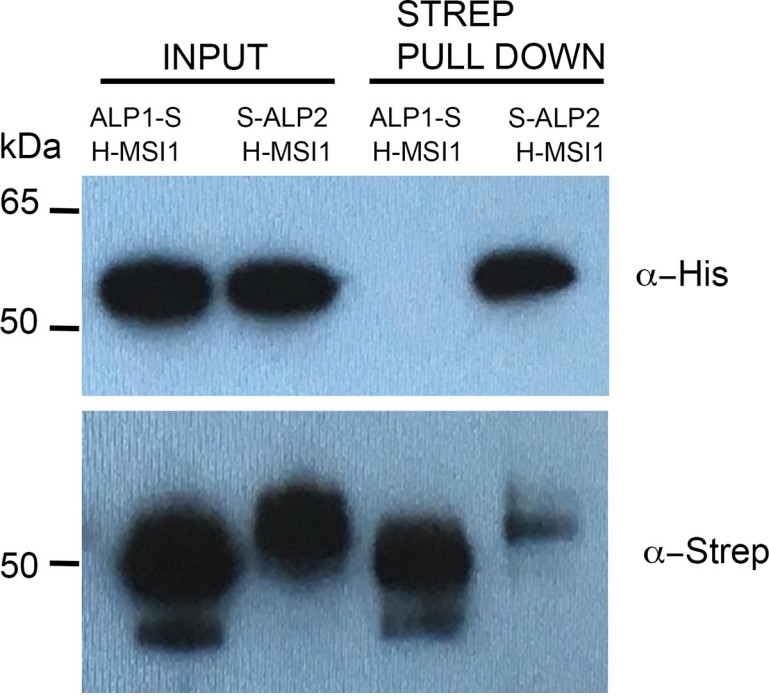

**Fig 7. Pull down analysis of ALP-PRC2 protein interactions.** The ALP1 or ALP2 proteins were co-expressed in insect cell as fusions with streptavidin binding peptides (ALP1-S, S-ALP2) together with MSI1 tagged with 6 X His (H-MSI1). The Strep-tagged ALP1 or ALP2 proteins were pulled down from whole cell extracts using Strepavidin beads and the eluted proteins analysed in immunoblots using antisera directed against the Strep and 6 X His tags, respectively. H-MSI1 was pulled down with ALP2 but not with ALP1.

Although full-length ALP1 and ALP2 proteins did not interact with any PRC2 member in yeast (S6 Fig), an N-terminal portion of ALP2 interacted with MSI1 as a bait but not a prey fusion (Fig 5C). As an additional confirmation of the ALP2 interaction with MSI1, we co-expressed epitope-tagged versions of ALP2 and MSI1 in insect cell cultures using a baculoviral expression system and performed pulldown assays. This showed that ALP2 could pull down MSI1, whereas in control experiments co-expressing ALP1 and MSI1 no interaction of ALP1 with MSI1 was found (Fig 7). Collectively, these experiments indicate that ALP2 but not ALP1 contacts the PRC2 via the MSI1 core member.

The above results suggested that the association of ALP1 with the PRC2 might be indirect and rely on ALP2 as a bridge. To test this, we first asked whether expressing ALP2 in tobacco could facilitate an interaction of ALP1 with MSI1 in BiFC assays. When tobacco leaves were co-infiltrated with a mixture of Agrobacterium cells harbouring *35S::ALP1-nYFP* and *35S::cYFP-MSI1* constructs, no fluorescence was observed. By contrast, when we included a third Agrobacterium strain harbouring *35S::ALP2*, many cells showed fluorescence indicating that ALP2 could cause ALP1 and MSI1 to associate closely with one another (Fig 6 and S7 Fig). To confirm that these interactions were relevant in Arabidopsis, we introduced an *pALP1::ALP1-GFP* transgene into the *alp2-1* mutant background and performed IP-MS. Although we identified numerous ALP1 peptides, indicating that ALP1 protein is expressed and stable in the *alp2* background, extremely few peptides from PRC2 members were detected (Table 1). In addition, a volcano plot analysis of replicate ALP1 IP samples showed that the core PRC2 were all highly enriched in the *ALP2*[+] background compared with *alp2-1*, consistent with ALP2 being necessary for ALP1 to interact with the PRC2 (Fig 4C and S3 Dataset). Together these

results indicate that ALP2 recruits ALP1 to PRC2 via direct interaction with the MSI1 component.

## Analysis of H3K27me3 enrichment in *alp* mutants

Since *ALP1* and *ALP2* acts antagonistically to the PRC2 by de-repressing PcG target genes (Fig 2), we analysed the effects on H3K27me3 deposition at target gene chromatin using chromatin immunoprecipitation (ChIP) from seedling extracts. We found that in *clf-28* mutants, consistent with previous studies [29, 39], H3K27me3 levels at selected PcG targets were decreased relative to the wild type, whereas the *alp* mutants showed no consistent change (Fig 8A and 8B). We next investigated whether the partial suppression of the *clf* phenotype in *clf alp* double mutants (Fig 1) might reflect a restoration of H3K27me3 at PcG targets, which may be caused, for example, by an increase in SWN-PRC2 activity when *ALP* function is removed. To this

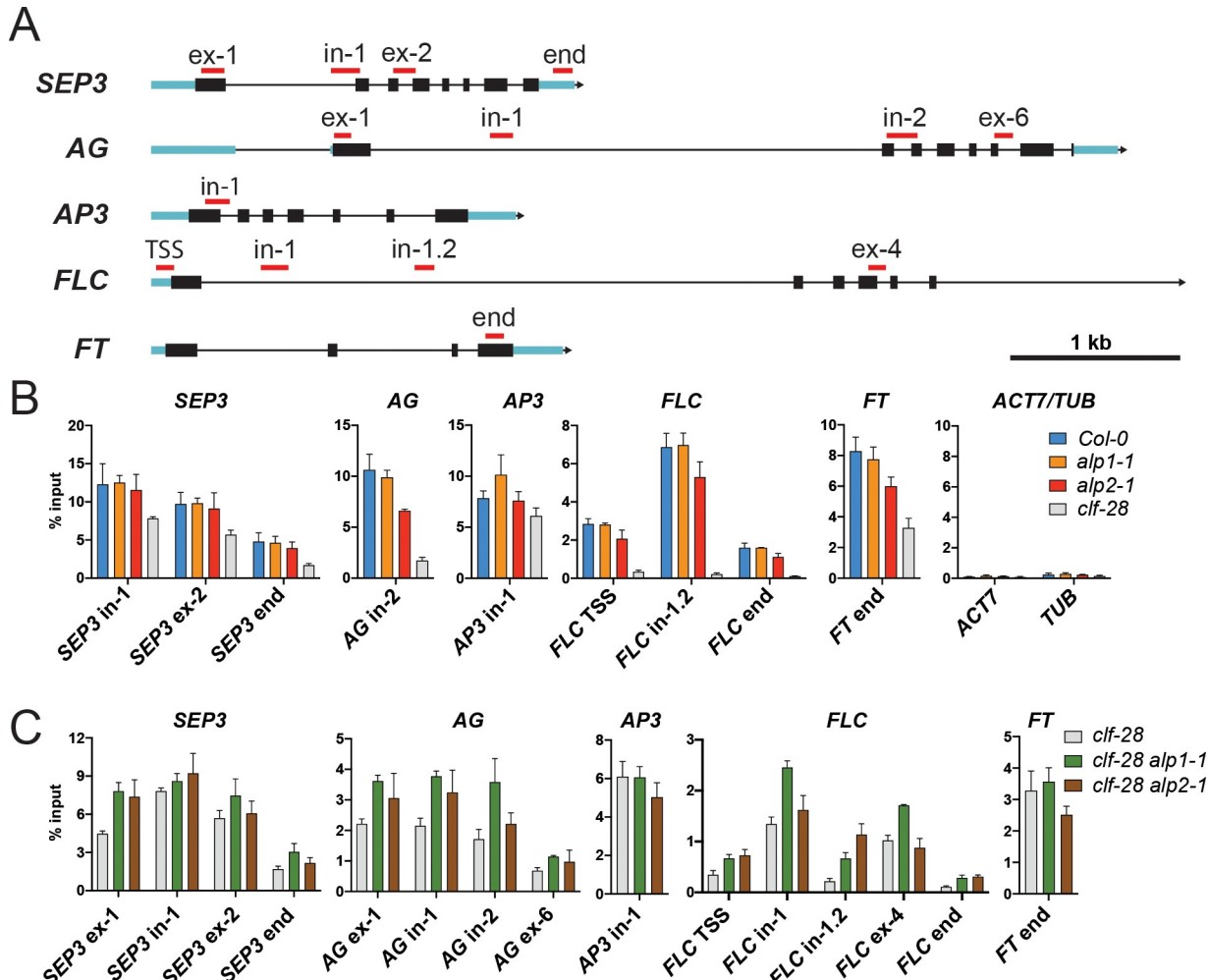

**Fig 8. ChIP analysis of H3K27me3 enrichment at loci known to be affected by PRC2-CLF activity.** A. Schematics of genomic regions examined during ChIP experiments. Approximate regions amplified via real-time PCR are denoted by red bars. Blue bars: untranslated regions; black bars: exons; black lines: introns; arrow heads: direction of transcription. B. Immunoprecipitation of chromatin prepared from 10-day-old Col-0, *alp1-1*, *alp2-1* and *clf-28* seedlings. An H3K27me3 antibody was used to precipitate chromatin, and enrichments are displayed as percentage input. C. Immunoprecipitation of chromatin from *clf-28*, *alp1-1 clf-28* and *alp2-1 clf-28* 10 day old seedlings. Three biological replicates were used for each genotype and error bars indicate the standard deviation from the mean. Primers that amplify regions in *ACT7* (AT5G09810) and *TUB2* (AT5G62690) were included as negative controls. In: intron; TSS: transcriptional start site.

end, we compared H3K27me3 levels in *clf-28*, *clf-28 alp1-1* and *clf-28 alp2-1* mutants by initially testing the same genomic regions used for Fig 8B and then additional regions for genes that appeared to show differences. We found that H3K27me3 levels at *SEP3*, *AG* and *FLC* were generally higher in the double mutants than in *clf-28* (Fig 8C), whereas little change was observed for *FT* and *AP3*. Thus, the incomplete suppression of the *clf* phenotype by *alp* mutations may be caused by a partial restoration of H3K27me3 deposition at some, but likely not all, PcG target genes.

## Discussion

We have shown that ALP1 and ALP2 have evolved from the nuclease and Myb DNA binding components of *Harbinger* transposases respectively, and that they are components of a variant PRC2 complex. The association with a histone methyltransferase complex regulating plant gene expression is striking since two other *Harbinger* related proteins have also been found to be part of a histone modifying complex in Arabidopsis [17]. These findings raise the question as to why former transposases have evolved associations with chromatin modifying complexes and why these domesticates have been conserved in flowering plants.

### ALP2 is related to the DNA binding component of *Harbinger* transposases

Whereas ALP1 shows similarity with *Harbinger* derived nucleases, the ALP2 protein sequence is more diverged and its evolutionary origins less apparent. Several observations however support that ALP2 has arisen by domestication of the Myb DNA binding component of a *Harbinger* transposase. Firstly, ALP2 protein interacts directly with the nuclease-derived ALP1, similar to the interaction of the two components of *Harbinger* transposases or of their domesticated derivatives such as Harb1/Naif in animals or HDP1/HDP2 in Arabidopsis [14, 17]. Secondly, the interaction is mediated by a C-terminal portion of ALP2, as has also been found for the nuclease interacting regions of HDP2, NAIF1 and *Ping* Myb [14, 17]. Lastly, alignment of plant ALP2 proteins with *Harbinger* transposases identify regions of similarity that include three regions previously shown to be conserved amongst plant *Harbinger* Mybs [13]. Importantly, the three tryptophan residues that are required for DNA binding activity and that are conserved in HDP2 and *Harbinger* Mybs are not conserved in ALP2, so it is unlikely to bind DNA [14, 17]. Similarly, two of the three residues (DDE) that are required for nuclease activity and are conserved in *Harbinger* nucleases have been lost in ALP1 [18, 35]. Collectively, these results suggest that ALP1 and ALP2 were domesticated from a plant *Harbinger* TE whose transposase components acquired mutations that disabled their ability to catalyse transposition. Once the two genes no longer encoded functional transposases, selection for the host to silence their expression will have been relaxed, and this may have allowed the ancestral ALP proteins to acquire novel interactions with host proteins as they continued to accumulate mutations and diverge (see Fig 9 for model). The fact that ALP2 has diverged more than ALP1 may reflect a general trend, as several other studies of plant and animal *Harbinger* transposases found the Myb DNA binding component to be less well conserved in sequence than the nuclease [11, 13].

### ALP2 recruits ALP1 to the PRC2 through interaction with MSI1

Our proteomic analyses provide a mechanistic basis for the genetic observation that ALP1 and ALP2 share mutant phenotypes and act in a common pathway. Thus both proteins are members of a variant PRC2 complex, ALP-PRC2. Independent assays–BiFC in tobacco, two hybrid in yeast, and pulldowns from insect cells–all indicate that ALP2 but not ALP1 interacts directly with the core PRC2 via MSI1, a WD repeat protein implicated in nucleosome binding. Because

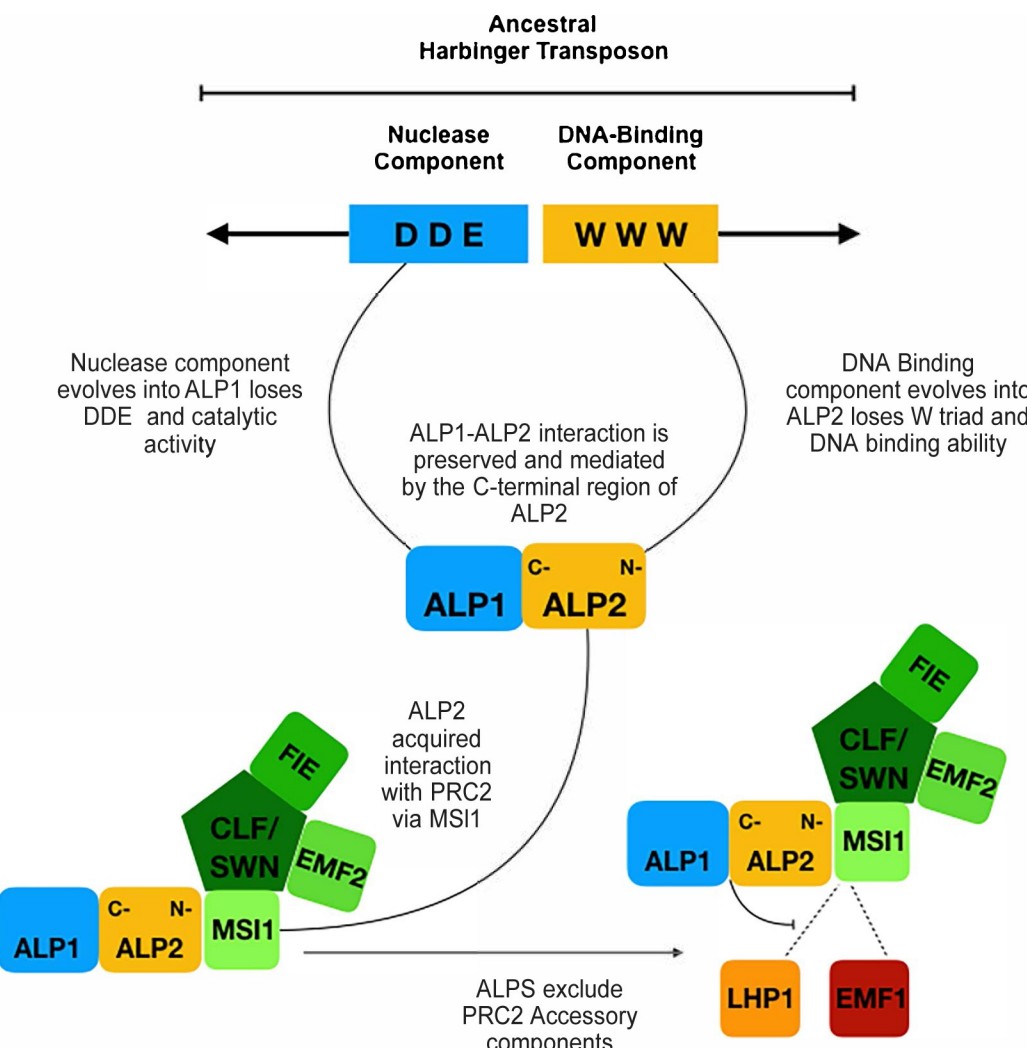

**Fig 9. Model for evolution of ALP-PRC2.** The ancestral transposases accumulated mutations that disabled their ability to catalyse transposition but maintained their interaction. The interaction of ALP2 with MSI1 displaced EMF1 and LHP1, which also interact with MSI1, leading to mutually exclusive PRC2 subcomplexes.

ALP2 can interact with both MSI1 and ALP1, via distinct N- and C-terminal regions respectively, it potentially recruits ALP1 to PRC2. Consistent with this, we found that only when ALP2 was co-expressed with ALP1-nYFP and cYFP-MSI1 in tobacco leaves was an ALP1-MSI1 interaction observed in BiFC assays, presumably because ALP2 brought the two proteins into close proximity. Additionally, immunoprecipitation experiments showed that *ALP2* activity was needed in order for ALP1 to associate with PRC2 components in Arabidopsis inflorescences, supporting that the interactions observed in the heterologous systems (tobacco and yeast) are biologically relevant.

## ALP1 and ALP2 form a distinct PRC2 complex

A distinctive feature of the ALP-PRC2 complex is that it lacks the accessory components that are commonly observed in pull down experiments involving core members of the PRC2. For example, IP-MS experiments variously using CLF, EMF2, MSI1 or FIE as baits have repeatedly identified EMF1, LHP1, VRN5 and VEL1 [18, 26, 29]. By contrast, IP-MS with ALP1 or ALP2

baits retrieved the four core PRC2 members but never these accessory components. It is notable that like ALP2, EMF1 and LHP1 were also found to interact directly with the MSI1 component of plant PRC2 [26, 40]. The ALP-PRC2 and the LHP1/EMF1-PRC2 may therefore form mutually exclusive subcomplexes via competition for access to PRC2 via MSI1, so that the ALP2/MSI1 interaction excludes LHP1 and EMF1 and *vice versa*. It currently unclear how VRN5 and VEL1 interact with the core PRC2, but proteomics data suggest that they copurify with core PRC2 but not EMF1 or LHP1, so may form a third PRC2 subcomplex [34].

The formation of mutually exclusive PRC2 subcomplexes via competition of accessory components for a core component has also been described in vertebrates where the PRC2.1 complex is distinguished by the presence of one of three Polycomb Like Proteins (PCL1-3) and either EPOP2 or PALI1/2, whereas PRC2.2 contains JARID2 and AEBP2 [for review see 41]. Structural studies and *in vitro* assays indicate that AEBP2 and PCL3 compete for binding to one region of the core component SUZ12, whereas JARID2 and EPOP2 compete for another region of SUZ12; in this way, the association of PRC2 with PCL and EPOP2 in PRC2.1 precludes formation of PRC2.2 and *vice versa* [24]. The different subcomplexes have overlapping and redundant roles in targeting the PRC2. Whereas the disruption of any one PRC2 subcomplex by inactivation of a specific accessory component had little effect, when PCL1-3 and JARID2 where all inactivated the core PRC2 subunits were no longer targeted to specific genes; global H3K27me3 levels were not greatly reduced, but the distribution of H3K27me3 changed so that it was dispersed at low levels genome wide and not focussed at specific targets [42, 43]. By analogy, it is possible that the role of ALP-PRC2 may overlap with that of other subcomplexes such as LHP1/EMF1-PRC2 and VRN5/VEL1-PRC2 so that the role of ALP activity may be more apparent when the other subcomplexes are inactivated.

## Role of ALPs in regulating PRC2 activity

The role of *ALP1* and *ALP2* is most apparent in backgrounds weakly compromised for PcG activity such as *lhp1* or *clf* mutants, where loss of *ALP* activity reduces the mis-expression of several key PcG targets and alleviates the mutant phenotypes. The simplest explanation is that association with ALPs reduce the H3K27me3 HMTase activity of PRC2, so that in a *clf* mutant background withdrawing *ALP* activity increases activity of the remaining SWN containing PRC2 and partially restores H3K27me3 levels. Consistent with this, we found a partial restoration of H3K27me3 at some CLF target genes in comparisons of *clf* and *clf alp* mutants (Fig 8C). The ALPs could inhibit SWN-PRC2 activity indirectly, by excluding accessory components which themselves promote activity, or more directly by interfering with catalytic activity. As an example of the latter, the EZHIP protein in humans associates with PRC2 and is thought to inhibit its HMTase activity by binding the active site of EZH2 [44].

By contrast, loss of *ALP* activity in an otherwise wild type PcG background has much less phenotypic effect and we observed no consistent alteration of H3K27me3 deposition at selected PcG targets in *alp* mutants relative to the wild type (Fig 8B). Similarly, our previous transcriptional profiling showed that the effects of *alp1* mutation on gene expression levels in a *CLF*+ background were much less than in a *clf* mutant background [18]. Despite this, the association of ALP proteins with the PRC2 occurs in wild-type backgrounds; in addition the *ALP* genes are widely conserved in flowering plants. The ALP proteins must therefore have important roles outside of PcG mutant backrounds. We previously found [18] that *alp1* mutants had reduced activation of *AP3* and *PI* floral homeotic genes which was revealed in weak mutant backgrounds for *LEAFY*, a transcription factor involved in *AP3*/*PI* activation and implicated in opposing PcG silencing by repression of *EMF1* expression [45]. Here, we show that *alp* mutants, particularly *alp2*, are slightly late flowering and exhibit delayed induction of floral

homeotic genes upon photoperiodic induction. The ALPs may therefore contribute to overcoming PcG repression during developmental transitions.

## Why are *Harbinger* transposons repeatedly domesticated

The Arabidopsis genome encodes at least four other *Harbinger* nuclease derived proteins with similarity to ALP1 and HDP1 (predicted products of *At3g55350*, *At5g12010*, *At4g29780*, *At3g19120*) and there are potentially others that are too diverged to be easily recognisable, as is the case for the Myb derived proteins such as ALP2 and HDP2. Since the two components of *Harbinger* transposases have also been co-domesticated in vertebrates and in Drosophila [11–14], this raises the question as to whether there are features of *Harbinger* class TE that predispose them for domestication. One distinctive feature is the modular structure of the transposase, in which the nuclease and DNA binding activities are encoded by separate proteins which interact with one another. During domestication the interaction between the two components has been preserved in all cases examined (Harbi1/Naif, HDP1/HDP2 and ALP1/ALP2) and similar regions of the proteins mediate the interactions. The domains correspond to the N-terminus of the nuclease and the C-terminus of the myb component in several functional *Harbinger* class TE including a reconstructed zebrafish *Harbinger* [14] and the *Ping* TE of rice (this study). Similar regions mediate the interaction in the domesticated forms, as HDP1 and HDP2 also interact via their N- and C-termini, respectively [17], and in ALP2 it is the C-terminus that mediates the interaction with ALP1. Although ALP1 and ALP2 interact with each other in yeast assays, they did not interact with the *Ping* nuclease and Myb proteins, suggesting that during domestication the interacting surfaces have co-evolved and diverged in the different lineages. Other features of the ancestral TE are less conserved, for example both HDP1 and ALP1 have lost the DDE triad required for nuclease activity, and ALP2 has also lost the tryptophans needed for Myb-mediated DNA binding. The modular nature of the *Harbinger* transposition complex may have facilitated the acquisition of novel interactions by the different components–for example, in HDP1/HDP2 the nuclease derived component interacts directly with IDM1, a component of a histone acetyl transferase complex, whereas in ALP1/ALP2 the Myb derived component interacts with MSI1.

A second feature of *Harbinger* TE is that they predominantly insert in single copy, gene rich regions of the genome [13, 46, 47]. It is striking that although the *Harbinger* domesticates have diverged from one another and from their ancestral transposases, they are well conserved throughout angiosperms. This conservation suggest that they are performing a function for the host, rather than promoting TE proliferation. Both the ALP and HDP proteins are associated with chromatin modifying complexes and in the case of HDP1/2 there is ChIP-seq and RNA-seq data to support that they regulate TE expression [17]. The association of domesticated *Harbinger* transposases with chromatin modifiers may therefore have been selected due to beneficial effects on host gene activity by targeting nearby TE. In the case of the two component *Harbinger* transposases, if one component acquired a novel interaction with a chromatin modifier but retained its ability to interact with its partner, this could help in recruitment of novel regulators–for example if the partner continued to bind TE sequences or to interact with other transposases that did so. One prediction of this model is that many other domesticated *Harbinger* transposases will also occur in chromatin modifying complexes and regulate host genes. It will be interesting to test this, for example by proteomic analysis of the numerous ALP1 related proteins encoded in the Arabidopsis genome.

## Materials and methods

### Plant materials

Plants were grown in controlled environment rooms at 21˚C under long day (16h light, 8h dark) or short day (8h light, 16h dark) growth conditions on shelves illuminated with

fluorescent strip lights on a compost mixture containing Levington's F2, Perlite, horticultural sand (in a ratio of 15:6:4). Plants for the IP-MS experiments were grown in controlled environment cabinets at 16°C in continuous light. All mutants were in Col-0 genetic background. The *alp1-1* and *alp2-1* mutants were identified in the genetic screen described in [35], the *alp2-2* mutation is a T-DNA insertion (Salk_150532) in the first exon of *ALP2*. The *pALP1::ALP1-GFP* and *35S::ALP1-GFP* transgenic lines were described previously [18].

## Total RNA extraction, cDNA synthesis and Gene Expression Analysis

Total RNA was extracted from 0.1 g of plant material, ground to fine powder in $N_2(l)$, using RNeasy Plant Mini Kit (Qiagen # 74904) according to the manufacturer's instructions, including the steps of on-column DNA digestion with RNase-Free DNase (Qiagen # 79254). First strand cDNA was synthesised using 1 μg total RNA primed with an anchored oligo dT primer and Superscript III reverse transcriptase (Thermo Fisher # C18080044-X) as described in product's documentation. Quantification of gene expression by real time PCR was performed using a LightCycler480 (Roche) and PCR reactions containing 1X SYBR master premix (Roche) as described in [18]. Crossing point (Ct) values were determined using the 2nd derivative max method in the LightCycler480 (Roche) software. For each primer set, the amplification efficiency was checked by analysis of a calibration curve made using a ten-fold dilution series ($10^0$–$10^{-3}$) of cDNA again using the supplied software. The crossing point (Ct) values were converted to relative expression or enrichment levels manually in Excel using the $2^{-\Delta\Delta Ct}$ method. Gene expression was normalised relative to the *EiF4A1* reference gene. The primers used to amplify target genes are listed in S3 Table.

## Generation of transgenic lines

To make the *35S::GFP-ALP2* construct, the *ALP2* coding sequence was amplified from first strand cDNA prepared from mRNA of seedlings of Ws ecotype using primers containing 5'-attB sequences and a proofreading polymerase (Phusion, New England Biolabs). The PCR product was introduced into the Gateway entry vector pDONR207 by recombination using BP clonase (Invitrogen) and the *ALP2* sequence confirmed by Sanger dideoxy-sequencing using the Edinburgh University Genepool facility. The *35S::GFP-ALP2* construct was generated by Gateway recombination using LR clonase to transfer the *ALP2* coding sequence into the Gateway destination vector pGWB6 [48]. To make the genomic *ALP2* construct, a 2kb fragment spanning the *ALP2* locus was amplified from genomic DNA of Col-0 plants using *att*B tailed primers and introduced into Gateway entry vector pDONR207 and then the destination vector pGDB2 as above. The constructs were introduced directly into Col-0 and *alp2-1 lhp1-3* genetic backgrounds by *Agrobacterium*-mediated floral dip transformation [49].

## Sequence retrieval, phylogenetic and synteny analysis

The ALP2 protein sequence was used in BLASTP and TBLASTN searches to identify the most similar proteins from diverse land plants. The retrieved proteins were used in reverse BLASTP searches against the Arabidopsis genome and in all cases retrieved the original query protein as the best hit. Where genome sequences were available, the sequences were retrieved by interrogating the Genbank and Phytozome databases. In other cases, the transcript sequences were obtained by querying the OneKP (one thousand plant genomes) database [50]. The retrieved sequences were aligned using MUSCLE [51] implemented within the Geneious package (https://www.geneious.com) and then alignments manually edited. For the phylogenetic analysis, sequences were aligned using MAFFT implemented in Geneious, the alignments manually trimmed, then gaps stripped using the stripped alignments function in Geneious (positions

with 40% or more gaps were removed). The phylogenetic trees were produced using the Mr Bayes program implemented on the CIPRES server (https://www.phylo.org/) with a mixed amino acid model, substitution rates modelled using invgamma ngamma = 4. The trees were run for up to 10 million generations 25% of trees discarded in the burn in phase. Convergence was defined as when the average split deviation frequency was less than or equal to 0.01. The output tree was imported and edited in Geneious. Synteny analysis was performed using the program Synteny Viewer available on the Arabidopsis Information Resource (TAIR). In addition, the syntenic regions were further confirmed manually using TBLASTN to retrieve the most similar proteins to those neighbouring ALP2 in *Arabidopsis thaliana*.

## Immunoprecipitation and mass spectrometry (IP-MS) of Inflorescence extracts

IPs were performed using 1-3g of inflorescence tissue harvested from plants grown in cabinets at 18˚C under 23 hours light: 1 hour dark cycles. Proteins were extracted in two volumes of buffer ((10mM Tris pH 7.5, 150 mM NaCL, 0.5% Igepal, 1% Triton) containing 1X protease inhibitor cocktail (Roche) and 0.1mM PMSF or 1mM Pefabloc (Roche). The IP was performed as described earlier [18] using GFP-trap agarose beads (Chromotek). Proteins were separated by NuPAGE Novex 4–12% Bis-Tris gel, (Life Technologies, UK), in NuPAGE buffer (MES) (Life Technologies, UK) and visualised using InstantBlue stain (Sigma Aldrich, UK). The stained gel bands were excised and de-stained with 50mM ammonium bicarbonate (Sigma Aldrich, UK) and 100% (v/v) acetonitrile (Sigma Aldrich, UK) and proteins were digested with trypsin, as previously described (Shevchenko et al. 1996). Briefly, proteins were reduced in 10 mM dithiothreitol (Sigma Aldrich, UK) for 30 min at 37˚C and alkylated in 55 mM iodoacetamide (Sigma Aldrich, UK) for 20 min at ambient temperature in the dark. They were then digested overnight at 37˚C with 12.5 ng μL-1 trypsin (Pierce, UK). Following digestion, samples were diluted with equal volume of 0.1% TFA and spun onto StageTips as described by [52]. Peptides were eluted in 40 μL of 80% acetonitrile in 0.1% TFA and concentrated down to 1 μL by vacuum centrifugation (Concentrator 5301, Eppendorf, UK). Samples were then prepared for LC-MS/MS analysis by diluting them to 5 μL with 0.1% TFA. LC-MS-analyses were performed on a Q Exactive Mass Spectrometer (Thermo Fisher Scientific, UK) and and on an Orbitrap Fusion™ Lumos™ Tribrid™ mass spectrometer (Thermo Fisher Scientific, UK), both coupled on-line, to an Ultimate 3000 RSLC nano Systems (Dionex, Thermo Fisher Scientific, UK). Peptides were separated on a 50 cm EASY-Spray column (Thermo Fisher Scientific, UK) assembled in an EASY-Spray source (Thermo Fisher Scientific, UK) and operated at a constant temperature of 50˚C. Mobile phase A consisted of 0.1% formic acid (Sigma Aldrich, UK) in deionised water while mobile phase B consisted of 80% acetonitrile and 0.1% formic acid. Peptides were loaded onto the column at a flow rate of 0.3 μL min-1 and eluted at a flow rate of 0.2 μL min-1 according to the following gradient: 2 to 40% mobile phase B in 90 min, then to 95% in 11 min and returned at 2% 6 min after. For the Rhino-ALP2 project the gradient was slightly different; 2 to 40% mobile phase B in 150 min, then to 95% in 11 min. For Q Exactive, FTMS spectra were recorded at 70,000 resolution (scan range 350–1400 m/z) and the ten most peaks with charge ≥ 2 of the MS scan were selected with an isolation window of 2.0 Thomson for MS2 (filling 1.0E6 ions for MS scan, 5.0E4 ions for MS2, maximum fill time 60 ms, dynamic exclusion for 60 s). Only ions with charge between 2 and 6 were selected for MS2. The normalized collision energy for the HCD fragmentation [53] that was used was set at 27. For the Orbitrap Fusion™ Lumos™, survey scans were performed at resolution of 120,000 (scan range 350–1500 m/z) with an ion target of 4.0e5. MS2 was performed in the Ion Trap at a rapid scan mode with ion target of 2.0E4 and HCD fragmentation with normalized collision

energy of 27 (Olsen *et al*, 2007). The isolation window in the quadrupole was set at 1.4 Thomson. Only ions with charge between 2 and 7 were selected for MS2. The MaxQuant software platform [54] version 1.6.6.0 was used to process raw files and search was conducted against the complete *Arabidopsis thaliana* database (Uniprot, released October 2016), using the Andromeda search engine [54]. The first search peptide tolerance was set to 20 ppm while the main search peptide tolerance was set to 4.5 pm. Isotope mass tolerance was 2 ppm and maximum charge to 7. Maximum of two missed cleavages were allowed. Carbamidomethylation of cysteine was set as fixed modification. Oxidation of methionine and acetylation of the N-terminal as well as the Gly-Gly (diglycil) on lysine were set as variable modifications. For peptide and protein identifications FDR was set to 1%. For the volcano plot statistical analysis, the spectra obtained from the mass spectrometric analysis of the Trypsin-digested IP-MS experimental samples were searched against the *Arabidopsis thaliana* proteome database (UniprotKB_aratha downloaded on 15.03.2015) and label free quantification performed using the MaxQuant programme [55] and the following search parameters: Digestion mode = Trypsin; Variable modifications = Acetyl (protein N terminus), Oxidation (M); Maximum missed cleavages = 4. The output of this analysis was analysed by a two sample T-test using Perseus software [56] to find proteins whose increased abundance in a given sample relative to the negative control (e.g. in sample expressing GFP-tagged protein of interest relative to the sample expressing GFP alone) was statistically significant.

The mass spectrometry proteomics data have been deposited to the ProteomeXchange Consortium via the PRIDE partner repository with the dataset identifier PXD018911

## Tandem Affinity Purification (TAP) and MS of GS$^{rhino}$-ALP2

The constructs expressing *35S::GSrhino-ALP2* were generated by Gateway recombination and PSB-D suspension cultures were transformed as previously described [38]. The tandem affinity purification followed the protocol described in [38] but the subsequent in gel tryptic digest and MS analysis was performed as described above for inflorescence extracts. Volcano plot analysis was performed on MaxQuant analysed date using the package DEP in R studio.

## Yeast two hybrid analysis

The constructs for yeast two hybrid assays with full length ALP1 and ALP2 proteins were generated by Gateway recombination using the destination vectors pGADT7-DEST (prey fusions) and pGBKT7-DEST (bait fusions) and ALP entry clones made in pDONR207 (as described above) and the *Ping* constructs were made in pENTR or pDONR Zeo by cloning PCR products as described previously (Hancock et al. 2010). The *alp1-1* fusions (G273E) were generated by site directed mutagenesis using the Quikchange kit (Stratagene). The constructs expressing the Arabidopsis PRC2 components were a generous gift from Dr Daniel Schubert (Free University of Berlin). The various ALP1 and ALP2 truncations were generated by PCR amplification of the relevant coding sequences using primers incorporating restriction enzyme sites, the products were first cloned as blunt end fragments using pJET vector (Fermentas) and the inserts then subcloned as in-frame fusions into pGADT7 AD and pGBKT7 vectors (Clontech) by conventional restriction enzyme digestion and ligation. Preparation and transformation of Y2HGold (Clontech) yeast-competent cells was carried out as described in [57]. Ten distinct large colonies (2–4 mm) were picked from each plate of transformed yeast and were pooled together in a 30 ml universal tube containing 5 ml of liquid Yeast Minimal Medium (YMM) enriched with CSM-L-W dropout supplement (Anachem # 4520–012). Universal tubes were then incubated overnight at 30˚C with rigorous shaking (250 rpm). On the next day, the OD$_{600}$ value of each liquid culture was measured and 1 ml was harvested by spinning down

the cells in a bench-top microcentrifuge for 30 sec at maximum speed. The supernatant was removed and the cells were washed by resuspending in 1 ml of sterile $dH_2O$. Cells were then pelleted again (30 sec max speed) and resuspended in adequate volume of 1x TE buffer, calculated from the $OD_{600}$ value of the original culture, so that for the resulting yeast suspension $OD_{600} = 1$.

Two 1/10 serial dilutions of every yeast suspension were prepared and 8 μl were immediately spotted on the different plates along with 8 μl of the undiluted sample. Non selective YMM + CSM-L-W plates were used as a control for the viability of the spotted yeast. Low stringency YMM + CSM-L-W-H (Anachem # 4530–112) plates were used to detect weak interactions and high stringency YMM + CSM-L-W-H-A (Anachem # 4540–412) plates, with or without Aureobasidin (Clontech # 630466), were employed for detection of stronger interactions.

### BiFC assay

The constructs for expression of *CLF* and *SWN* in BiFC assays were described previously[28]. For *FIE* the coding sequences were PCR-amplified with Gateway-compatible specific primers and introduced into the Gateway entry vector pDONR221 by BP clonase mediated recombination. For *MSI1* and *EMF2* constructs, Gateway entry clones containing full length coding sequences (accessions G25372 and U16887) were obtained from the Arabidopsis Biological Resource Center (ABRC). For *ALP1*, *alp1-1* and *ALP2* the Gateway entry clones generated for yeast two hybrid analysis (see above) were used. In all cases, the coding sequences were transferred into the split YFP binary destination vectors pBATL-B (for split YFP fusions at C-terminus) and pCL112/3 (for split YFP fusions at N-terminus) [58] by LR mediated Gateway recombination. For expression of untagged ALP2, a *35S::ALP2* construct was generated by Gateway recombination using the ALP2 entry clone and the destination vector pGWB2 [48]. *Agrobacterium tumefaciens* strains carrying plasmids for BiFC were grown overnight at 28 ˚C in 10 mL selective YEP medium until reaching $OD_{600} = 1{,}7–2$. Cells were then collected by centrifugation, and resuspended in appropriate volume of infiltration medium (10 mM MgCl2, 10 mM MES-KOH, pH 5.6) so that $OD_{600} = 0.3–0.4$. The bacterial suspensions with the BiFC constructs to be tested were combined and left to rest on the bench for 3 h. Subsequently, cells were infiltrated into the abaxial surface of 3-week-old *Nicotiana benthamiana* plants. The YFP fluorescence signal was observed and recorded with a Leica SP8 confocal laser-scanning microscope. Specimens were examined with the 10x objective (overview) or the 40x water-immersion objective (detailed pictures), using the OPSL 488 laser for excitation.

### Protein expression in insect cells using Baculovirus and pull down assays

The construct for expressing 6XHis-tagged MSI1 was described previously [59] and was provided by Nicolas Thoma (Friedrich Miescher Institute for Biomedical Research, Switzerland). For generation of the ALP1-SSF construct, pFASTIN-FS vector (a modified version of pFasT-Bac [ThermoFisher] kindly provided by D. Reinberg, New York University) was digested with *Nco*I and *Xho*I. The *ALP1* coding sequence (CDS), amplified using primers with suitable restriction endonuclease extensions and excluding the stop codon, was inserted into the pFAS-TIN-FS backbone as an in-frame translational fusion with the Dual Strep Flag tag (SSF) via conventional restriction enzyme digestion / ligation reactions. SSF-ALP2 was constructed via a Gibson Assembly strategy. Briefly, the pFASTIN vector was first digested with *Nco*I and *Xho*I, which removes the HIS tag and the multiple cloning site (MCS). The SSF tag and the ALP2 CDS were amplified from the pFASTIN-FS vector and cDNA, respectively, using primers bearing suitable overlapping regions for assembling an SSF-ALP2 translational fusion into

the pFASTIN vector backbone. Primers were designed using the online assembly tool http://nebuilder.neb.com/#!/ and the assembly itself was performed with the NEBuilder HiFi DNA Assembly Cloning Kit (NEB # E5520S) following the suggested protocols. The recombinant pFASTIN constructs bearing the ALPs-SSF fusions were verified by sequencing and then introduced into DH10Bac cells by heat shock transformation. Putative transformants (white colonies) were selected (blue/white selection) on Bluo-Gal (Thermo Fisher Scientific) containing LB agar plates supplemented with appropriate antibiotics. Recombinant bacmids were isolated from putative transformants via standard miniprep procedure and successful integration of the ALPs-SSF fusions was confirmed via PCR.

*Spodoptera frugiperda* Sf9 cells were grown in suspension cultures in HyClone CCM3 medium (GE # SH30065.02) at 27°C and 125 rpm (1" orbit). Cell density was maintained between 0.5–6x10$^6$ cells/ml. To generate baculovirus stocks, 1.8x10$^6$ Sf9 cells were plated in a 35-mm tissue culture dish and transfected with 1μg of recombinant bacmid using X-treme-GENE HP DNA Transfection Reagent (Roche # 06 366 236 001) following the manufacturer's instructions. After 4–5 days, cell culture supernatants containing recombinant baculovirus particles were clarified by centrifugation at 1000 x g for 5 min at room temperature and used for further titer amplification in suspension cultures at a density of 2x10$^6$ cells/ml, using initially a 1:20 dilution of the clarified supernatant, followed by 2–3 sequential rounds of amplification with 1:100 dilutions of the previous virus stock.

For protein expression and Pull-Down assays, Sf9 cells at a density of 4x10$^6$ cells/ml were inoculated with the highest titer baculovirus stocks at a dilution of 1:60 each of either 6xHIS-MSI1 and ALP1-SSF, or 6xHIS-MSI1 and SSF-ALP2. After 65 h, cells were harvested via centrifugation at 500 x g for 5 min at room temperature. The pellet was either frozen in LN$_2$ and stored at -80 $^o$C or used immediately for protein extraction. Total protein was extracted by resuspending the cell pellet in 8 ml of pre-chilled lysis buffer (150 mM NaCl, 10% v/v Glycerol, 50 mM Tris HCl pH 8, 0.1% v/v Triton-X-100, 1 mM DTT, Protease inhibitors cocktail tablets, 1 mM EDTA). The cell suspension was subjected to three 30-s sonication cycles, 35% amplitude, 1 s Pulse ON and 0.1 s Pulse OFF, and subsequently centrifuged at 40.000 x g for 30 min at 4 $^o$C. Supernatant was recovered and filtered through a MillexHA 0.45 μM syringe filter unit. 100 μl were set aside as an input sample and were immediately treated with DTT-containing 2X laemmli buffer. The rest of the cleared extract was subjected to Strep PullDown using the Gravity Flow Strep-Tactin XT Superflow 0.2 ml columns (IBA # 15543077) as per manufacturer's instructions.

## Chromatin immunoprecipitations for qPCR assays

Seedlings were grown on 0.5X MS-agar for 10 days at 16°C under constant illumination.

Whole seedlings were harvested, washed in dH2O, and dried with paper towels before freezing in liquid Nitrogen [N$_2$(l)] and storage at -80°C. For each chromatin extraction, 1 g of seedlings (fresh weight) was crosslinked in 1X crosslinking buffer (Diagenode) supplemented with 1% formaldehyde (Sigma 252549) at room temperature for 15 minutes under vacuum. Glycine was added to a final concentration of 0.125 M and a vacuum was applied for a further 5 minutes. The tissue was washed with 40 ml dH2O three times, dried with paper towels, and frozen in liquid Nitrogen [N$_2$(l)]. The tissue was ground to fine powder using a pestle and mortar [pre-cooled with N$_2$(l)] and incubated in 30 ml ice-cold extraction buffer 1 (Diagenode) supplemented with 0.1X protease inhibitors (Sigma P9599) for 5 minutes with gentle rocking at 4°C. The homogenate was filtered twice through Miracloth (Calbiochem) into pre-cooled tubes and centrifuged at 3000 x g for 20 minutes at 4°C. The pellet was suspended thoroughly in 1 ml ice-cold extraction buffer 2 (0.4 M sucrose, 100 mM Tris-HCl pH 8.0, 1%

Triton X-100, 10 mM MgCl2, 0.1 mM PMSF, 5 mM β-ME, 0.1X protease inhibitors) and centrifuged at 12,000 x g for 10 minutes at 4˚C. The pellet was washed and centrifuged once more with extraction buffer 2 and the supernatant was fully removed. The pellet was suspended in 300 μl ice-cold extraction buffer 3 (1.7 M sucrose, 100 mM Tris-HCl pH8.0, 0.15% Triton X-100, 2 mM MgCl2, 0.1 mM PMSF, 5 mM β-ME, 0.1X protease inhibitors), transferred gently onto 300 μl extraction buffer 3 inside a 1.5 ml centrifuge tube, and centrifuged at 12,000 x g for 10 minutes at 4˚C. The supernatant was removed and the pellet was suspended in 300 μl sonication buffer (Diagenode) and incubated on ice for 5 minutes. The nuclei suspension was sonicated using a Diagenode Bioruptor Pico for 7 cycles of 30 seconds on/off at 4˚C, and centrifuged at 12,000 x g for 5 minutes at 4˚C. The supernatant was frozen in $N_2(l)$ and stored at -80˚C until use.

ChIP assays for each genotype were performed in triplicate and in parallel. Chromatin was thawed on ice and centrifuged at 12,000 x g for 5 minutes at 4˚C to pellet insoluble material. The supernatant was diluted 5-fold in 1 X ChIP Dilution Buffer (CDB; Diagenode). Samples (200 μl diluted chromatin) were incubated with rotation at 4˚C for 1 hour with 1 μg of αH3K27me3 (Millipore 07–449) or Rabbit IgG (Diagenode). 4 μl diluted chromatin was taken as input. 20 μl DiaMag protein A-coated magnetic beads were used per ChIP. The beads were washed 3 times in 200 μl CDB and suspended in their original volume before addition to the ChIP tube. Samples were incubated with the beads at 4˚C with rotation for 1 hour. The beads were washed once with each of wash buffers 1/2/3 (Diagenode) and twice with wash buffer 4 before suspension in 100 μl elution buffer 1 (Diagenode) and incubation with shaking at 65˚C for 15 minutes. 4 μl elution buffer 2 was added to each of the supernatants (separated from the beads) and incubated overnight at 65˚C with shaking. ChIP DNA was purified from the eluates using the iPure kit v2 (Diagenode) and used for qPCR.

ChIP-qPCR was performed using a Roche LightCycler 480 and SYBR Green I Master in a final volume of 10 μl. % Input values were calculated using the following formula: % Input = (100/Df) x [2^(Cp Input–Cp Sample)], where Df: dilution factor; Cp: crossing point PCR cycle. At least 2 technical replicates were performed for each sample.

## Immunoblots

For immunoblotting, proteins separated by SDS-PAGE were transferred onto nitrocellulose membrane using iBlot2 NC mini stacks (Invitrogen # IB23002) and the semi-dry Gel Transfer Devise iBlot2 (Invitrogen # IB21001). A 3-step transfer program was applied: 20V for 1 min, 23V for 4 min and 25V for 2 min. Following transfer, membranes were stained with Ponceau S solution (0.1% w/v Ponceau S, 1% v/v acetic acid) to demonstrate equal loading of protein samples. Subsequently, membranes were immersed for 1 h in blocking solution (5% w/v non-fat dried milk in TBS-T, 25 mM Tris-HCl pH 8, 150 mM NaCl, 2.7 mM KCl, 0.05% v/v Tween-20) to prevent non-specific binding of the antibodies. Primary antibodies were used in working dilutions ranging from 1:1000 to 1:5000 with the incubation time varying from 1 h at room temperature to overnight at 4˚C. Membranes were then washed 3 times with TBS-TT (25 mM Tris-HCl pH 8, 150 mM NaCl, 2.7 mM KCl, 0.1% v/v Triton-X, 0.05% v/v Tween) and once with TBS-T for a total of 20 min. Depending on the organism in which the primary antibodies had been produced, secondary anti-rabbit (GE healthcare # NA9340V) or anti-mouse (Cell Signalling Technologies # 7076) HRP conjugated antibodies, were used in 1:5000 and 1:1000 dilutions respectively, all prepared in blocking solution. The incubation time was 1 h followed by five washes with TBS-TT and 1 wash with TBS for a total of 35 min. For detection of Chemiluminescent signals, ECL Plus Western Blotting Detection Reagent (Pierce Fisher # 32132) was used according to the manufacturer's instructions.

## Supporting information

**S1 Fig. Induction of flowering genes in day-shifted plants.** RT-qPCR of *SEP3*, *AG*, and *PI* in SAM-enriched tissue of Col-0, *alp1-1*, and *alp2-1* plants that were grown under short day conditions (8 hr light, 16 hr dark) for 14 days, and then grown under long day conditions (16 hr light, 8 hr dark). Tissue was collected on the days specified after the photoperiod shift 1 hr prior to the commencement of the dark period (ZT 15). Values are the mean of three biological replicates and are presented relative to the reference gene *PP2AA3* (*AT1G13320*). Error bars indicate the standard deviation of the mean
(PDF)

**S2 Fig. Phylogeny of ALP2, HDP2 and Harbinger transposases.** An unrooted Bayesian phylogenetic tree was made based on alignment of the amino acid sequences. Numbers at the branches indicate probabilities (percent), the scale bar indicates the average number of substitutions per site. ALP2 sequences are indicated in red, *Harbinger* transposases in blue and HDP2 sequences in green.
(PDF)

**S3 Fig. Synteny between *A. thaliana* and *A. lyrata* in *ALP2* region.** In *A. thaliana*, *ALP2* is neighboured by *At5G24490* and *At5G24510* encoding 30S and 60S ribosomal proteins, respectively. The *A. lyrata ALP2* orthologue (*Al ALP2*) is located in a syntenous region, i.e. is flanked by orthologous genes with the same arrangement and orientation as in *A. thaliana*.
(PDF)

**S4 Fig. ALP2-GFP protein expression.** Immunoblot analysis of total protein extracts using α-GFP antibodies. A protein with the predicted size (65.2 Kda) for the ALP2-GFP fusion is specifically detected in extracts from *35S::GFP-ALP2* plants. *35S::LHP1-GFP* is included as a positive control.
(PDF)

**S5 Fig. The core PRC2 co-purifies with GS^rhino^-ALP2.** Volcano plot analysis with the x axis showing relative abundance of proteins identified in tandem affinity purified *35S:GSrhino-ALP2* samples relative to non transgenic control. The y axis shows probability values. Analysis is based on three biological replicates. ALP proteins highlighted in red, PRC2 components in blue.
(PDF)

**S6 Fig. Yeast two hybrid assays for ALP protein interactions.** A. Truncated forms of ALP1 did not interact with ALP2. The interaction of a C-terminal region of ALP2 (residues 171–261) is included as a control. B. Full length ALP1 and ALP2 proteins did not interact with core PRC2 components.
(PDF)

**S7 Fig. ALP protein interactions in BiFC assays.** Low magnification images showing epidermal cells from *N. benthamiana* leaves transformed by infiltration with Agrobacterium. Images in left of panels are YFP channel, on right is merge of light field and fluorescence channels. ALP proteins are tested with each other and with the core PRC2 components. Interactions were only seen for the ALP1-ALP2 and MSI1-ALP2 combinations.
(PDF)

**S8 Fig. Interaction of Ping and ALP proteins in yeast two hybrid assays.** The full length *Ping* nuclease and Myb DNA binding proteins interact reciprocally as bait and prey fusions. No interaction above background was found between ALP and *Ping* proteins. Serial ten-fold

dilutions of five pooled transformants were spotted onto selective media.
(PDF)

**S9 Fig. Interaction of truncated Ping and ALP proteins in yeast two hybrid assays.** A N-terminal fragment (amino acids 1–223) of the Ping nuclease interacts with the full length Ping Myb DNA binding protein reciprocally. A C-terminal fragment (amino acids 292–465) of the Ping Myb DNA binding protein interacts with the Ping nuclease as a bait but not as a prey fusion. No interaction was found between truncated ALP proteins and Ping proteins. Serial ten-fold dilutions of five pooled transformants were spotted onto selective media.
(PDF)

**S1 Table. Identification of *ALP2* candidate genes by sequencing.** A. Annotation of high scoring mutations from fast isogenic mapping identified two candidate genes (Hartwig et al. 2012). Score indicates Shore quality score ranged from 1 (low) to 40 (high). Protein position indicates amino acid number in translated coding sequence. B. Allele frequency at the two candidate mutations reveals high frequency of the mutant allele in DNA from bulked mutant plants.
(PDF)

**S2 Table. IP-MS results using *35S::GS^rhino^-ALP2* in suspension cells.** Table shows the number of uniquely identified peptides from ALP proteins, core PRC2 subunits and accessory components. Three replicate experiments. Non transgenic PSB-D Arabidopsis suspension culture cells were used as control.
(PDF)

**S3 Table. Oligonucleotide primer sequences.** Excel sheet providing the sequences and rationale for the different primers described in materials and methods section.
(XLSX)

**S1 Text. ALP2, HDP2 and Harbinger protein sequences.** Text file with the species names, accession numbers and protein sequences used for the alignment in Fig 2 and the phylogenetic tree in S2 Fig. Sequences were mostly retrieved from Genbank, other than where indicated *Harbinger* transposase sequences which were retrieved from Repbase and the gymnosperm and pteridophyte sequences which were retrieved from OneKP.
(DOCX)

**S1 Dataset. Results from yeast two hybrid screen using ALP1 bait construct.** Excel sheet showing the identity of putative interacting proteins determined by Sanger sequence analysis of prey constructs retrieved from yeast colonies growing on -LWHA plates. Two colonies harbouring C terminal fragments of the ALP2 protein were found (rows 397 and 435).
(XLSX)

**S2 Dataset. Full datasets for IP-MS and TAP-MS experiments.** Excel sheets for three replicate experiments, for each sample the number of uniquely identified peptides mapping to a specific protein is indicated.
(XLSX)

**S3 Dataset. Volcano plot analysis of relative protein abundance.** Excel sheet with results from statistical analysis of MS data. Protein abundance was determined using label free quantification (LFQ) in Maxquant, and the statistical analysis (two sample t test) performed using Perseus. FDR and S0 values were adjusted to filter the samples most highly enriched and significant (indicated by a + sign).
(XLSX)

## Acknowledgments

We thank Pat Watson, Billy Adams and Sophie Haupt for care and maintenance of plants, Matthew Lyst for advice on IP-MS, Brendan Davies and Barry Causier for advice on yeast two hybrid assays. Weronika Borek and Kasia Rataj gave helpful advice on volcano plot analysis of mass spectrometry data.

## Author Contributions

**Conceptualization:** Franziska Turck, Frank Wellmer, Justin Goodrich.

**Data curation:** Christos N. Velanis, Christos Spanos, Franziska Turck, Justin Goodrich.

**Formal analysis:** Christos N. Velanis, Pumi Perera, Bennett Thomson, Serin Gümüs, Christos Spanos, Franziska Turck, Justin Goodrich.

**Funding acquisition:** Frank Wellmer, Justin Goodrich.

**Investigation:** Christos N. Velanis, Pumi Perera, Bennett Thomson, Erica de Leau, Shih Chieh Liang, Ben Hartwig, Harry Thornton, Pedro Arede, Jiawen Chen, Kimberly M. Webb, Serin Gümüs, Christos Spanos, Juri Rappsilber, Franziska Turck, Justin Goodrich.

**Methodology:** Christos N. Velanis, Pumi Perera, Shih Chieh Liang, Ben Hartwig, Alexander Förderer, Juri Rappsilber, Philipp Voigt, Franziska Turck, Frank Wellmer, Justin Goodrich.

**Project administration:** Erica de Leau, Frank Wellmer.

**Resources:** Alexander Förderer, Geert De Jaeger, Clinton A. Page, C. Nathan Hancock, Juri Rappsilber, Philipp Voigt, Franziska Turck, Frank Wellmer.

**Supervision:** Christos N. Velanis, Pumi Perera, Erica de Leau, Kimberly M. Webb, Juri Rappsilber, Philipp Voigt, Franziska Turck, Frank Wellmer, Justin Goodrich.

**Writing – original draft:** Franziska Turck, Justin Goodrich.

**Writing – review & editing:** Christos N. Velanis, Pumi Perera, Frank Wellmer, Justin Goodrich.

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
