## [Decision Letter · Decision Letter 0]

20 Aug 2019

Dear Dr Goodrich,

Thank you very much for submitting your Research Article entitled 'The ALP domesticated transposases antagonise Polycomb group gene function in plants and interact with a core subunit of Polycomb Repressive Complex 2 (PRC2)' to PLOS Genetics. Your manuscript was fully evaluated at the editorial level and by independent peer reviewers. The reviewers appreciated the attention to an important problem, but raised some substantial concerns about the current manuscript. Based on the reviews, we will not be able to accept this version of the manuscript, but we would be willing to review again a much-revised version. We cannot, of course, promise publication at that time.

As you will see from the detailed comments of the reviewers, they both consider your work interesting and important and appreciate the functional insight into ALP2 function. However, both have suggestions and queries that should be addressed before publication, and we agree that a few additional experiments and modified statements would strengthen the publication. We kindly ask you to pay special attention to the questions of the ALP2 role in a wt background and the extension of the interaction analysis, suggested by both reviewers. We think that this is likely to give valuable additional hints to the mechanistic connection and the evolution of the TE “domestication”. The minor comments can all be considered by changing or shortening the text accordingly. We hope that you will find them helpful during the revision.

If you decide to revise the manuscript for further consideration at PLOS Genetics, please aim to resubmit within the next 60 days, unless it will take extra time to address the concerns of the reviewers, in which case we would appreciate an expected resubmission date by email to plosgenetics@plos.org.

[LINK]

We are sorry that we cannot be more positive about your manuscript at this stage. Please do not hesitate to contact us if you have any concerns or questions.

Yours sincerely,

Ortrun Mittelsten Scheid

Associate Editor

PLOS Genetics

Wolf Reik

Section Editor: Epigenetics

PLOS Genetics

Reviewer's Responses to Questions

**Comments to the Authors:**

Reviewer #1: This is a review of the manuscript entitled “The ALP domesticated transposase antagonize Polycomb group gene function in plants and interact with a core subunit of the Polycomb Repressive Complex 2 (PRC2)” submitted to PLoS Genetics. Overall, I found this manuscript very interesting and exciting, with the understanding that although I do not work on transposable element domestication, I find this entire field fascinating. In this work, the authors identify the second Harbinger family transposable element that is able to interact with host histone post-translational modification and therefore gene regulation. The strength of this manuscript is the proteomics (IP-Mass Spec) and protein-protein interaction data. The weakness of the manuscript was a lack of ChIP-seq for H3K27me. I’ve split my comments into major points and minor changes.

Major points

1. The overall function of the ALP proteins was never detected in wild-type or single mutant alp plants, but could only be seen in an already-quite severe lhp chromatin mutant background (although the single alp2 mutant has a phenotype – Figure 1A). Therefore, I think the authors need to reduce their claims of ALP protein function, because they don’t really know this role in normal plants (if they have any role at all). I suggest softening the overall conclusions regarding ALP protein function, including in the abstract and title of the manuscript.

a. The obvious experiment missing here is H3K27me ChIP-seq in wt and the ALP mutants. This experiment would define what role the ALP proteins have in a non-lhp mutant chromatin context.

2. The introduction reads too long, and I don’t think the example of SPM and MU elements that have their methylation changed by the presence of the transposase is a good example of opposing host silencing. In this case, methylation is likely responding to the binding of the transposase protein, and not a direct interference of a host silencing mechanism. There are other examples of direct repression of host silencing that would be more suitable, especially from viruses.

3. HP1 is a very well-studied protein in animals. The authors need to add a sentence or two to explain to the reader how LHP1 is different in plants (does not bind H3K9me).

4. The ALP1-ALP2 proteins interact, and the interaction domain was mapped in Figure 5. Is this domain necessary for interaction that same domain responsible for Harbinger TE nuclease/DNA binding protein interaction? Is this the interaction that the host has reused / domesticated, even though the DNA binding is gone?

a. What aspects of Harbinger transposons do the authors think lead to their repeated domestication? It seems from their data that it may be the protein-protein interaction. I ask the authors to please speculate on this point in the Discussion section.

Minor changes to be made

1. The authors define “domestication” as a process whereby the transposase proteins encoded by a TE acquire novel host functions. However, can’t domestication function on other TE proteins, such as GAG, Pol, etc…? I favor a more inclusive definition whereby domestication refers to any reuse of a TE for host function, including TE DNA sequences as binding sites and enhancer elements.

The introduction calls Arabidopsis TEs inactive, but I think this is a misnomer. They are inactive in wt somatic tissues of the reference strain, but active in the species (see eLife 2016;5:e15716).

2. The first section of the results is very long. I suggest reducing the verboseness, and creating a new paragraph at “To characterize these phenotypes further…”.

3. I did not understand why Figure 2B was important, or where it was referred to in the Results section. This could be moved to the Supplementary data or discussed more in the main text.

4. Figure 2A is really important to the manuscript. What is the control gene for the qRT-PCR that was used?

Reviewer #2: The manuscript by Perera et al entitled “The ALP domesticated transposases antagonize Polycomb group gene function in plants and interact with a core subunit of Polycomb Repressive Complex 2 (PRC2)” suggests that a group of evolutionarily coopted transposons controls an epigenetic regulator complex which mediates the H3K27me3 in Arabidopsis. This study is essentially an extension of the authors’ previous work of ALP1 but adds considerably novel discovery (ALP2 and functional roles). Although the work is interesting and of broad readership to the field, it seems the manuscript requires some improvement before publication. Here are my comments:

1. Arguments without supporting evidence should be removed. For example,

“as we had independently identified this protein in a yeast two hybrid screen for ALP1 interactors (data not shown)”

2. One of the key data in this study is that ALP2 acts as a bridging protein for ALP1 and PRC2 core. Although the authors presented BiFC and IP-MS data supporting this, I would like to suggest that authors perform additional independent experiments to strengthen the argument. E.g. co-IP shown as western blot.

3. Relating the ChIP analyses in Figure 8, it would be nice if ALP1 and ALP2 binding to the target is shown. Plus, I believe ChIP-seq can be carried out with no big difficulties.

4. A schematic model illustrating the work would be good to be included.

**Have all data underlying the figures and results presented in the manuscript been provided?**

Reviewer #1: Yes

Reviewer #2: Yes

PLOS authors have the option to publish the peer review history of their article (what does this mean?). If published, this will include your full peer review and any attached files.

Reviewer #1: No

Reviewer #2: No

---

## [Decision Letter · Decision Letter 1]

18 Feb 2020

Dear Dr Goodrich,

We are pleased to inform you that your manuscript entitled "The domesticated transposase ALP2 mediates formation of a novel Polycomb protein complex by direct interaction with MSI1, a core subunit of Polycomb Repressive Complex 2 (PRC2)" has been editorially accepted for publication in PLOS Genetics. Congratulations!

Yours sincerely,

Ortrun Mittelsten Scheid

Associate Editor

PLOS Genetics

Wolf Reik

Section Editor: Epigenetics

PLOS Genetics

Comments from the reviewers (if applicable):

Reviewer's Responses to Questions

**Comments to the Authors:**

Reviewer #1: The authors have completed the revision in a satisfactory manner.

Reviewer #2: The revised version of the manuscript is significantly improved in its quality by supplementing additional data and description as per reviewers' suggestions. I do not have any objection to the acceptance of this manuscript to publication.

**Have all data underlying the figures and results presented in the manuscript been provided?**

Reviewer #1: Yes

Reviewer #2: Yes

PLOS authors have the option to publish the peer review history of their article (what does this mean?). If published, this will include your full peer review and any attached files.

Reviewer #1: No

Reviewer #2: No

**Data Deposition**

http://datadryad.org/submit?journalID=pgenetics&manu=PGENETICS-D-19-01239R1

**Press Queries**

---

## [Editor Report · Acceptance letter]

19 May 2020

PGENETICS-D-19-01239R1 

The domesticated transposase ALP2 mediates formation of a novel Polycomb protein complex by direct interaction with MSI1, a core subunit of Polycomb Repressive Complex 2 (PRC2) 

Dear Dr Goodrich, 

We are pleased to inform you that your manuscript entitled "The domesticated transposase ALP2 mediates formation of a novel Polycomb protein complex by direct interaction with MSI1, a core subunit of Polycomb Repressive Complex 2 (PRC2)" has been formally accepted for publication in PLOS Genetics! Your manuscript is now with our production department and you will be notified of the publication date in due course.

With kind regards,

Kaitlin Butler

PLOS Genetics

On behalf of:
